

# Anisotropic scaling of the two-dimensional Ising model I: the torus

**Hendrik Hobrecht[1] and Alfred Hucht[1⋆]**

1 Fakultät für Physik, Universität Duisburg-Essen, Lotharstr. 1, D-47048 Duisburg, Germany

⋆ fred@thp.uni-due.de

## Abstract

We present detailed calculations for the partition function and the free energy of the finite two-dimensional square lattice Ising model with periodic and antiperiodic boundary conditions, variable aspect ratio, and anisotropic couplings, as well as for the corresponding universal free energy finite-size scaling functions. Therefore, we review the dimer mapping, as well as the interplay between its topology and the different types of boundary conditions. As a central result, we show how both the finite system as well as the scaling form decay into contributions for the bulk, a characteristic finite-size part, and – if present – the surface tension, which emerges due to at least one antiperiodic boundary in the system. For the scaling limit we extend the proper finite-size scaling theory to the anisotropic case and show how this anisotropy can be absorbed into suitable scaling variables.

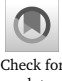
# 1 Introduction

The two-dimensional Ising model on the square lattice is by far the most examined and best understood non-trivial system in statistical physics. Albeit there are still open questions, there is a whole plethora of properties known exactly, starting with the exact partition function on the torus in the thermodynamic limit calculated by Onsager [1,2], over the universal finite-size scaling at its continuous phase transition from a ferromagnetic low-temperature to a paramagnetic high-temperature phase [3, 4], to the exact solutions at criticality due to conformal field theory for arbitrary geometries and boundary conditions (BCs) [5–8]. There are as many ways to calculate all these properties as there are people working on the topic, but a few are truly worth mentioning here. Beneath Onsager's expansion of the transfer matrix calculation to two dimensions, there are at least two major ideas, which are repeatedly used over last decades; one is the mapping onto spinors, as done lately by Baxter for the case of a rectangular geometry with open boundaries [9] and originally introduced by Kaufman [2]. The other one is the dimer mapping, introduced by Kasteleyn [10–12], refined by Fisher [13], and rigorously and exhaustively examined by McCoy & Wu [14–16], see [17] for a comprehensive presentation of the topic. Only quite recently, a connection between these two methods was established [18, 19], which reduces the Pfaffians emerging in the dimer method to matrices corresponding to the spinor picture even for arbitrary couplings, therefore preserving the possibility to apply arbitrary boundary conditions in both directions [20]. Note that this correspondence goes beyond the simpler case of translational invariant couplings, where both methods are known to lead to the same $2 \times 2$ matrices. The dimer mapping will be our starting point, and we will be putting a lot of effort into its analysis relating the topology of the underlying graph and the boundary conditions of the spin system.

In this work we will focus on the finite-size contributions to the free energy in systems with periodic BCs in both directions and arbitrary couplings in one direction. This is in contrast to the works on the infinitely extended "layered Ising model" [16, 21, 22], were the configuration of couplings varies within a given number of layers and repeats periodically, without any finite-size contributions. In our analysis we will separate the leading contribution of the thermodynamic limit and analyse its first finite-size correction, which is responsible for the critical Casimir effect, and calculate the corresponding finite-size scaling functions. Boundaries that destroy the translational invariance in one direction as well as additional surface fields were discussed in the literature for the case of the half plane [14, 15, 23] and the slab geometry [24–27]; its generalisation to the cylinder with finite aspect ratios will be the topic of a subsequent paper [28].

The original inspiration for this work lies within the aforementioned universal finite-size scaling; in the vicinity of criticality the behaviour of many systems can be sorted into associated universality classes, categorised only by some rough properties like its spatial and spin dimensions, and split up into subcategories treating the BCs confining the system. Accompanying the universality, Fisher & de Gennes [29, 30] predicted that the diverging correlation length $\xi^{(\infty)}$ at criticality gives, in finite systems, rise to the thermodynamic analogue of the quantum-electrodynamic (QED) Casimir effect [31, 32]. The critical fluctuations present in such a geometrically confined system lead to effective forces between the systems surfaces, namely the critical Casimir forces. In contrast to the always attractive QED Casimir effect, these thermal forces may be attractive or repulsive depending on the BCs of the surfaces, and can even change their behaviour with the temperature. This fact together with its steering temperature dependence makes them especially interesting as experimental model systems, e. g., they may be used to control the interaction strength in colloidal suspensions in order to investigate the aggregation processes [33–36]. The effect itself was first measured by Garcia & Chan as a critical thinning of a $^4$He film near the $\lambda$-transition [37]. The universality of

this measurement was first proven by Monte Carlo simulations of the $XY$-model in three dimensional thin films with Dirichlet boundary conditions [38]. Other experiments were made on thin films of $^3$He-$^4$He mixtures near the tricritical point [39], again in the $XY$-universality class, and binary liquids, whose demixing transition is in the Ising-universality class [40]. The latter experimental system was expanded to the direct measurement of interactions between spherical particles and a potentially chemically striped surface as well as the observation of aggregation processes in the above-mentioned colloidal suspensions [41].

The analytical point of view of those systems is either restricted to mean-field calculations [42], the large-$n$ limit [43, 44], renormalization group theory [45–52], or calculations at criticality [53–60], where the systems conformal invariance opens up some fascinating possibilities [61–64]; furthermore there are many works on Monte Carlo simulations of such systems [38, 65–67]. Starting with the original slab geometry, where some sort of thermodynamic integration is necessary to obtain the free energy, they were lately expanded to finite aspect ratios [52, 68] as well as to spherical objects [69–72], which establishes the opportunity to implement the same protocols for a direct measurement of the free energies due to the probability distributions of the positions of the objects.

For the present work, in Section 2 we first recapitulate the mapping between the two-dimensional Ising model and closest-packed dimers on a carefully chosen lattice as proposed by Kasteleyn, and expand the argumentation to arbitrary nearest-neighbour couplings between each pair of sites. Therefore we will refine the argumentation by McCoy & Wu and, in contrast to the calculations for the rectangular system with open boundary conditions, we will implement translational invariance in one direction to simplify the problem further as done in [16], which leaves us with the toroidal and the cylindrical topology, where the latter one will be the topic of a subsequent paper [28]. Allowing anisotropic couplings, we will show how the introduction of dual couplings leads to a much more natural calculation compared to one of our previous works [73]. Later on, the general calculation of the emerging determinant gives us the opportunity to implement several different boundary conditions, but also makes a recapitulation of the scaling theory necessary, especially for the anisotropic case, which will be done in Section 3. Here we will also recapitulate the relation between the different points of view concerning the preferred direction within the system, e. g., the direction in which the critical Casimir force is measured, as well as the relations between the scaling functions for the free energy and the critical Casimir force.

Afterwards we will start with the calculation of the partition function for the anisotropic toroidal case in Section 4, which will be a crucial point for the calculation for the symmetric (++) and the antisymmetric (+−) boundary fields in [28], as it enlightens the interdependency of the dimer mapping and the distinction of periodic and anti-periodic boundary conditions. Additionally it gives us a good opportunity to identify the bulk contribution, which is present in all cases examined later on. We apply the anisotropic scaling theory to the Onsager dispersion relation in order to obtain the scaling forms of the relevant terms and introduce its hyperbolic parametrisation, which allows us to regularise the infinite sums by rewriting them as contour integrals over singly periodic functions in the complex plane, giving a modified Abel-Plana formula, and reproduce the free energy scaling function for the toroidal case [68, 73, 74]. With at least one antiperiodic boundary the system forms a domain wall, which introduces a surface tension contribution to the residual free energy, giving us a first glimpse on its general form. The cylindric geometry and boundary field effects will be discussed in [28].

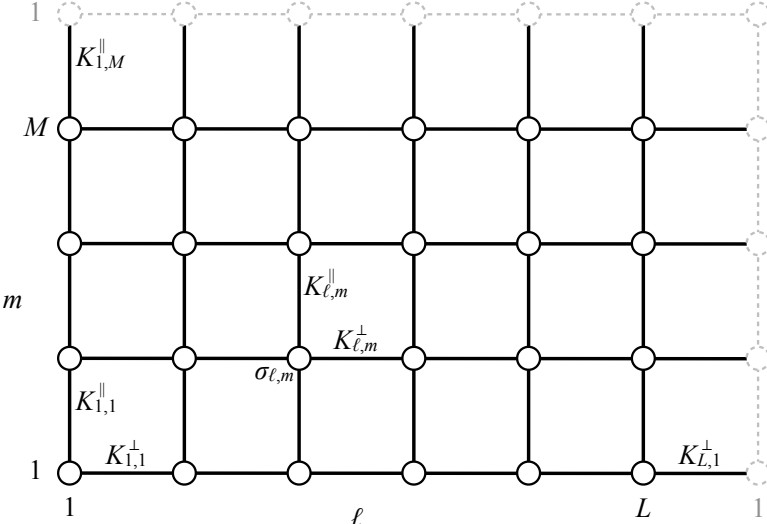

Figure 1: The square lattice with toroidal geometry for $M = 4$ and $L = 6$.

## 2 The dimer representation

We will start with a brief summary of the dimer representation of the two-dimensional Ising model as it was introduced by Kasteleyn [10,11] and refined by Fisher [13] and McCoy & Wu [16], as well as an explanation of how the corresponding matrices are constructed. Then we will assume translational invariance in one direction and reduce the calculation of the determinant successively by two Schur reductions. This makes it convenient to introduce the dual couplings, which lead to a more readable and natural form. Eventually we obtain the determinant of a (quasi-)cyclic tridiagonal matrix, which can be calculated in terms of a simple $2 \times 2$ transfer matrix. This step will be our starting point for all further calculations, as the various BCs can be simply introduced in the reduced matrix and thus form such $2 \times 2$ matrices for the boundary terms. Finally we assume translational invariance in both direction, reproducing the result of McCoy & Wu.

The two-dimensional Ising model on a finite $L \times M$ square lattice with periodic boundary conditions (p) in both directions as depicted in Fig. 1, i. e., on the torus, is described by the reduced Hamiltonian (in units of $k_{\mathrm{B}}T$, with Boltzmann constant $k_{\mathrm{B}}$)

$$\mathcal{H}^{(\mathrm{p},\mathrm{p})} = -\sum_{\ell=1}^{L}\sum_{m=1}^{M} K_{\ell,m}^{\perp}\sigma_{\ell,m}\sigma_{\ell+1,m} + K_{\ell,m}^{\parallel}\sigma_{\ell,m}\sigma_{\ell,m+1}, \tag{2.1}$$

where $K_{\ell,m}^{\perp}$ and $K_{\ell,m}^{\parallel}$ are the reduced couplings between the nearest neighbours in perpendicular and parallel direction, respectively, and $\sigma_{\ell,m} \in \{-1,+1\}$ are spin variables with periodic indices, i. e., $\sigma_{\ell+L,m} \equiv \sigma_{\ell,m} \equiv \sigma_{\ell,m+M}$. The partition function $Z^{(\mathrm{p},\mathrm{p})} = \mathrm{tr}\, e^{-\mathcal{H}^{(\mathrm{p},\mathrm{p})}}$ can be rewritten into a high-temperature expansion [75]

$$\frac{Z^{(\mathrm{p},\mathrm{p})}}{Z_0^{(\mathrm{p},\mathrm{p})}} = \frac{1}{2^{LM}}\sum_{\{\sigma\}}\prod_{\ell=1}^{L}\prod_{m=1}^{M}\left(1 + z_{\ell,m}^{\perp}\sigma_{\ell,m}\sigma_{\ell+1,m}\right)\left(1 + z_{\ell,m}^{\parallel}\sigma_{\ell,m}\sigma_{\ell,m+1}\right), \tag{2.2}$$

with $z_{\ell,m}^{\delta} = \tanh K_{\ell,m}^{\delta}$ ($\delta = \perp, \parallel$) and with the non-singular part

$$Z_0^{(\mathrm{p},\mathrm{p})} = \prod_{\ell=1}^{L}\prod_{m=1}^{M} 2\cosh K_{\ell,m}^{\perp}\cosh K_{\ell,m}^{\parallel}. \tag{2.3}$$

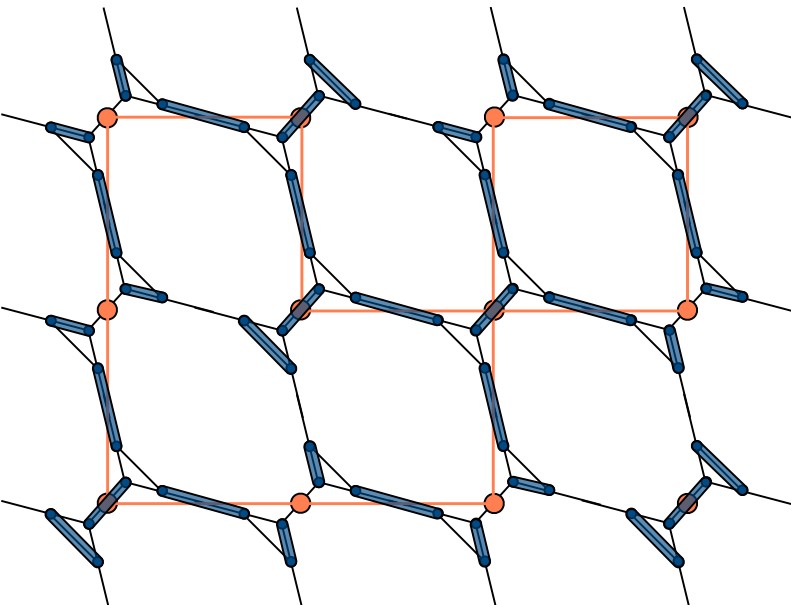

Figure 2: Example for the dimer representation of the graphical interpretation of the partition function on a $(L = 4) \times (M = 3)$ lattice. The original lattice sites are depicted as large circles, while the orange lines are one example for a polygon configuration. All eight possible dimer configurations can be seen. The corresponding term in the sum in (2.2) is $z^{\perp}_{1,1} z^{\perp}_{2,1} z^{\perp}_{2,2} z^{\perp}_{3,2} z^{\perp}_{1,3} z^{\perp}_{3,3} z^{\parallel}_{1,1} z^{\parallel}_{1,2} z^{\parallel}_{2,2} z^{\parallel}_{3,1} z^{\parallel}_{3,2} z^{\parallel}_{4,2}$.

Here we used the identity

$$e^{K\sigma_i \sigma_j} = \cosh K + \sigma_i \sigma_j \sinh K, \tag{2.4}$$

as $\sigma_i \sigma_j = \pm 1$ depending on whether the spins are aligned or unaligned.

Expanding the sum and both products leaves only even powers of the $\sigma_{\ell,m}$, which are always equal to one, and a factor of $2^{LM}$, which we already included in $Z_0^{(\mathrm{p,p})}$. This can be interpreted as follows: Each term in the sum gives a possible configuration of closed and commonly intersecting polygons on the original lattice, where every $z_{\ell,m}^{\perp/\parallel}$ gives the position of a perpendicular/parallel link that has to be drawn, see Fig. 2 for an example.

### Dimers

If we replace each site of the lattice with a properly chosen cluster, this problem is equivalent to finding the generating function of the closest-packed dimer configuration on the expanded lattice [10, 11, 14]. In this context a dimer is a two-atomic construct, which always occupies two sites and the connecting bond of the lattice. Fisher introduced a six-site cluster for this replacement [13], see Fig. 3(a), which can be further reduced to a four-site cluster [11], see Fig. 3(b). The dimers can be arranged in such a way that they reflect the polygon structure in a biunique way, occupying either an original lattice edge or lying within the cluster, thus leaving the edge unoccupied, see again Fig. 2.

The problem of finding the generating function of the closest-packed dimer configurations on an arbitrary planar graph was solved by Kasteleyn in terms of Pfaffians, as it gives the number of *perfect matchings* of a given directed planar graph with an even number of sites. This is especially powerful because of the connection between the Pfaffian and the determinant, namely

$$(\mathrm{Pf}\,\mathcal{A})^2 = \det \mathcal{A}. \tag{2.5}$$

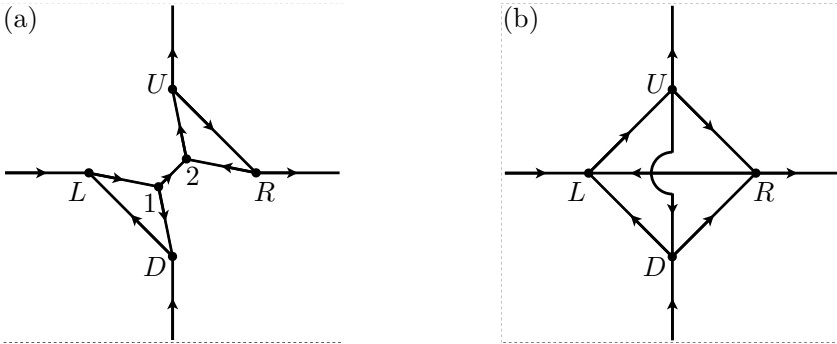

Figure 3: The six-site (a) and the four-site (b) cluster that may be used to expand the lattice for the dimer mapping.

To apply this method, we have to construct a suitable skew-symmetric matrix, which corresponds to the directed graph of the lattice; the construction of this matrix will be the topic of this section, were we choose a different way than the original works of Kasteleyn and Mc-Coy & Wu (c. f. chapter IV and section 2 of chapter V in [17]) and start with matrices with only positive entries and later antisymmetrise them. The directed graph we need has to fulfil some restrictions, the most important and fundamental being that each elementary polygon is *clockwise odd*, that is, the number of edges pointing in clockwise direction has to be odd. This is naturally satisfied for the cluster expanded lattice, and guarantees that every term in the Pfaffian will have the same sign on any given planar graph. Consequently, for the toroidal topology of the original lattice, one Pfaffian is not sufficient, since the argument aforementioned originates in the need that every *transitions cycle*, which describes the transition from one dimer configuration to another, needs to be odd, see [17, Chapter 4.3-4.5] for details. On a torus this cannot be accomplished by a single but by a superposition of four Pfaffians, corresponding to the four possible combinations of periodic and anti-periodic boundary conditions in the two directions, that cancels out the redundant terms and besides gives the correct signs for the transition cycles that wind around the torus in either or both directions [76]. These four combinations of periodic and anti-periodic BCs are implemented by reversing the direction of the directed graphs in the lines responsible for the periodicity. This last point will be discussed in detail in Section 4, where the interplay of the dimer mapping and periodic and anti-periodic boundaries will be discussed in more detail, as well as later on in the upcoming second part of this paper [28], where we will map the $(++)$ and the $(+-)$ boundary conditions onto periodic and anti-periodic BCs on a suitably extended lattice.

To construct the matrix we will first take a look at the adjacency matrix of the graph, which decomposes into the clusters replacing the original sites and the two direction of the lattice. Using the common labels for the cluster as shown in Fig. 3(a), the graph can be represented as

$$
C_6^0 = \begin{array}{c} \\ R \\ L \\ 1 \\ 2 \\ U \\ D \end{array}
\begin{array}{c} \begin{array}{cccccc} R & L & 1 & 2 & U & D \end{array} \\
\left( \begin{array}{cccccc}
0 & 0 & 0 & 1 & 0 & 0 \\
0 & 0 & 1 & 0 & 0 & 0 \\
0 & 0 & 0 & 1 & 0 & 1 \\
0 & 0 & 0 & 0 & 1 & 0 \\
1 & 0 & 0 & 0 & 0 & 0 \\
0 & 1 & 0 & 0 & 0 & 0
\end{array} \right) \end{array} .
\tag{2.6}
$$

To simplify the problem further, the graph can be reduced into a non-planar one with only four sites, eliminating the two interior sites labeled 1 and 2, e. g., as the Schur complement (2.25)

of the antisymmetric form of (2.6) with respect to the central $2 \times 2$ block. The related cluster is depicted in Fig. 3(b), and its adjacency matrix is

$$
C_4^0 = \begin{array}{c} \\ R \\ L \\ U \\ D \end{array} \overset{\begin{array}{cccc} R & L & U & D \end{array}}{\begin{pmatrix} 0 & 1 & 0 & 0 \\ 0 & 0 & 1 & 0 \\ 1 & 0 & 0 & 1 \\ 1 & 1 & 0 & 0 \end{pmatrix}}. \tag{2.7a}
$$

Nevertheless, this reduction introduces two additional terms for each lattice site of the original system that is not part of any polygon, as there are now three combinations on the Kasteleyn cluster for those cases. But if each bond within the clusters has a weight equal to 1, those additional terms cancel each other and we ensure that no cluster has an influence on (2.2). On this resolution, the coupling in parallel and perpendicular direction are represented by the two matrices

$$
C_4^{\parallel} = \begin{array}{c} \\ R \\ L \\ U \\ D \end{array} \overset{\begin{array}{cccc} R & L & U & D \end{array}}{\begin{pmatrix} 0 & 1 & 0 & 0 \\ 0 & 0 & 0 & 0 \\ 0 & 0 & 0 & 0 \\ 0 & 0 & 0 & 0 \end{pmatrix}} \quad \text{and} \quad C_4^{\perp} = \begin{array}{c} \\ R \\ L \\ U \\ D \end{array} \overset{\begin{array}{cccc} R & L & U & D \end{array}}{\begin{pmatrix} 0 & 0 & 0 & 0 \\ 0 & 0 & 0 & 0 \\ 0 & 0 & 0 & 1 \\ 0 & 0 & 0 & 0 \end{pmatrix}}, \tag{2.7b}
$$

respectively. The nearest-neighbour structure in a row and a column are both represented by the $n \times n$ matrix

$$
H_{b,n} = \begin{pmatrix} 0 & 1 & 0 & \cdots & 0 \\ 0 & 0 & 1 & & 0 \\ \vdots & & & \ddots & \vdots \\ 0 & 0 & 0 & & 1 \\ -b & 0 & 0 & \cdots & 0 \end{pmatrix}, \tag{2.8}
$$

with $b \in \{+1, 0, -1\}$ accounting for different BCs, i.e., $b = 0$ for open, $b = +1$ for periodic, and $b = -1$ for anti-periodic boundaries of the Ising system, concerning the necessity of transition cycles on the graph to be odd. Note, that $b = -1$ accounts for periodic boundaries on the directed graph, i.e., in the dimer system, in the sense that all edges are likewise aligned, while it accounts for antiperiodic BCs in the Ising model. However, the topology of the underlying directed graph is not representative for the Ising model, which is emphasised by the fact that the Ising partition function is a combination of four Pfaffians. The identification with periodicity and antiperiodicity for the values of $b$ stems solely from the identification within the open cylinder, see [28, 73], and thus we will use "+" to mark periodic and "−" to mark antiperiodic BCs. If we give each bond a different weight, see Fig. 1, we can represent the parallel and perpendicular couplings by

$$
\mathcal{Z}_{\alpha}^{\perp} = Z^{\perp} \left( H_{\alpha,L} \otimes \mathbf{1}_M \right), \tag{2.9a}
$$

$$
\mathcal{Z}_{\beta}^{\parallel} = Z^{\parallel} \left( \mathbf{1}_L \otimes H_{\beta,M} \right), \tag{2.9b}
$$

with $Z^{\delta} = \mathbf{diag}(z_{1,1}^{\delta}, z_{1,2}^{\delta}, \ldots, z_{L,M}^{\delta})$ and the $n \times n$ identity matrix $\mathbf{1}_n$. The final $4LM \times 4LM$ adjacency matrix of the graph then reads

$$
A_{\alpha\beta} = C_4^0 \otimes \mathbf{1}_{LM} + C_4^{\perp} \otimes \mathcal{Z}_{\alpha}^{\perp} + C_4^{\parallel} \otimes \mathcal{Z}_{\beta}^{\parallel} \tag{2.10}
$$

and, since two vertices are connected at most by a single edge, we can expand it to a skew-symmetric block form[1]

$$\mathcal{A}_{\alpha\beta} = A_{\alpha\beta} - A_{\alpha\beta}^{\intercal} = \begin{bmatrix} \mathbf{0} & 1 + \mathcal{Z}_{\beta}^{\parallel} & -1 & -1 \\ -(1 + \mathcal{Z}_{\beta}^{\parallel})^{\intercal} & \mathbf{0} & 1 & -1 \\ 1 & -1 & \mathbf{0} & 1 + \mathcal{Z}_{\alpha}^{\perp} \\ 1 & 1 & -(1 + \mathcal{Z}_{\alpha}^{\perp})^{\intercal} & \mathbf{0} \end{bmatrix} \tag{2.11}$$

to calculate the partition function of the torus as

$$\frac{Z^{(\text{p,p})}}{Z_0^{(\text{p,p})}} = \frac{1}{2}\left(\text{Pf}\,\mathcal{A}_{++} + \text{Pf}\,\mathcal{A}_{+-} + \text{Pf}\,\mathcal{A}_{-+} - \text{Pf}\,\mathcal{A}_{--}\right). \tag{2.12}$$

**Translational invariance**

Since we are interested only in the boundary conditions along the parallel direction, we now assume translational invariance in this direction and thus present an alternative solution to the problem of the layered Ising model presented in [16]. In contrast to the original calculations we do not restrict ourselves to the cylindrical case but include it as the special case of the torus, with $\beta = 0$. The Ising model on the rectangle with open boundaries in both directions is discussed in [9, 18, 19] for all temperature and in [77, 78] for the critical case. To account for the translational invariance in parallel direction, all $M$ couplings in each column are the same, $z_{\ell,m}^{\delta} \equiv z_{\ell}^{\delta} \; \forall \; m$, and the matrices $\mathbf{Z}^{\delta}$ simplify to

$$\mathbf{Z}^{\delta} = \mathbf{z}_L^{\delta} \otimes \mathbf{1}_M, \qquad \delta = \parallel, \perp, \tag{2.13}$$

with $\mathbf{z}_L^{\delta} = \mathbf{diag}(z_1^{\delta}, \dots, z_L^{\delta})$. Additionally we note that $H_{\text{b}}$ is a normal matrix for $\text{b} = \pm 1$, therefore it commutes with its transposed,

$$H_{\pm} H_{\pm}^{\intercal} - H_{\pm}^{\intercal} H_{\pm} = \mathbf{0}, \tag{2.14}$$

and thus the unitary matrix $U_{\pm}$ that diagonalises $H_{\pm}$ also does so for $H_{\pm}^{\intercal}$. For convenience we decompose the matrix $\mathcal{A}_{\alpha\beta}$ into its three contributions (2.10) as

$$\mathcal{A}_{\alpha\beta} = \mathcal{A}^0 + \mathcal{A}_{\alpha}^{\perp} + \mathcal{A}_{\beta}^{\parallel}, \tag{2.15}$$

where

$$\mathcal{A}^0 = C_4^0 \otimes \mathbf{1}_L \otimes \mathbf{1}_M - (transposed), \tag{2.16a}$$

$$\mathcal{A}_{\alpha}^{\perp} = C_4^{\perp} \otimes \left(\mathbf{z}_L^{\perp} H_{\alpha,L}\right) \otimes \mathbf{1}_M - (transposed), \tag{2.16b}$$

$$\mathcal{A}_{\beta}^{\parallel} = C_4^{\parallel} \otimes \mathbf{z}_L^{\parallel} \otimes H_{\beta,M} - (transposed). \tag{2.16c}$$

In the expanded space, $U_{\beta=\pm}$ becomes

$$\mathcal{U}_{\beta} = \mathbf{1}_4 \otimes \mathbf{1}_L \otimes U_{\beta,M} \tag{2.17}$$

and has no influence on the first two contributions as they commute,

$$\mathcal{A}^0 \mathcal{U}_{\beta} - \mathcal{U}_{\beta} \mathcal{A}^0 = \mathbf{0}, \tag{2.18a}$$

$$\mathcal{A}_{\alpha}^{\perp} \mathcal{U}_{\beta} - \mathcal{U}_{\beta} \mathcal{A}_{\alpha}^{\perp} = \mathbf{0}. \tag{2.18b}$$

---

[1]In the following we drop the size subscript from matrices if it can be derived from the context.

For the parallel contribution, we obtain the diagonal matrices

$$U_\beta^{-1} H_\beta U_\beta = \mathbf{diag}\left(e^{+i\varphi_0^{(\beta)}}, \ldots, e^{+i\varphi_{M-1}^{(\beta)}}\right), \tag{2.19a}$$

$$U_\beta^{-1} H_\beta^\mathsf{T} U_\beta = \mathbf{diag}\left(e^{-i\varphi_0^{(\beta)}}, \ldots, e^{-i\varphi_{M-1}^{(\beta)}}\right), \tag{2.19b}$$

with

$$\varphi_m^{(\beta)} = \begin{cases} 2m\pi/M & \text{if } \beta = -1 \\ (2m+1)\pi/M & \text{if } \beta = +1 \end{cases} \tag{2.20}$$

for $m \in \{0, 1, 2, \ldots, M-1\}$, and thus we will call $\beta = -1$ *even* and $\beta = +1$ *odd*. Note that the eigenvalues lie equidistantly on the unit circle and thus we have a free shifting parameter for the spectrum. We have chosen it in such a way, that the eigenvalue $\varphi_0^{(-)} = 0$ appears in the even spectrum; a shift by $-\pi$ on the other hand would have given rise to a dependency on whether $M$ is even or odd. For completeness we present the characteristic polynomials

$$\mathcal{P}_\beta^\pm(M; \varphi) = \prod_{m=0}^{M-1}\left(e^{\pm i\varphi} - e^{i\varphi_m^{(\beta)}}\right) = e^{\pm iM\varphi} + \beta \tag{2.21}$$

here, as we will need them later, too. Note that we have chosen to use the normalised version of $\mathcal{P}_\beta^\pm$ to omit the aforementioned dependency on whether $M$ is even or odd.

Since $\det(A \otimes B) = \det(B \otimes A)$, we can rearrange the matrices such that the product is block diagonal, simplifying its determinant to the form

$$\det \mathcal{A}_{\alpha\beta} = \prod_{m=0}^{M-1} \det \mathcal{B}_\alpha(\varphi_m^{(\beta)}), \tag{2.22}$$

with the $4L \times 4L$ block matrices

$$\mathcal{B}_\alpha(\varphi_m^{(\beta)}) = \begin{bmatrix} 0 & J_\beta^+ & -1 & -1 \\ J_\beta^- & 0 & 1 & -1 \\ 1 & -1 & 0 & J_\alpha \\ 1 & 1 & -J_\alpha^\mathsf{T} & 0 \end{bmatrix}, \tag{2.23}$$

where we defined $J_\alpha = 1 + z^\perp H_\alpha$ for the perpendicular direction and the diagonal matrices $J_\beta^\pm = 1 \pm z^\parallel e^{\pm i\varphi_m^{(\beta)}}$ for the parallel one.

At this point we introduce the dual couplings $t_\ell \equiv (z_\ell^\parallel)^*$, which can be defined within the low-temperature expansion of the partition function analogous to (2.2), but on the dual lattice. They are connected to the couplings $z_\ell^\parallel$ via the self-inverse duality transform $x^*$ of an arbitrary quantity $x$,

$$x^* \equiv \frac{1-x}{1+x}, \qquad\qquad (x^*)^* = x. \tag{2.24}$$

Thus we will omit the superscripts in the following and write $z_\ell \equiv z_\ell^\perp$ and rewrite the parallel couplings $z_\ell^\parallel$ through their duals $t_\ell^*$. In the following we will, if necessary, mark a dual coupling by an asterisk as shown in (2.24). Consequently we may also rewrite the two matrices $z^\perp \mapsto z$ and $z^\parallel \mapsto t^*$.

As shown in [73] the determinant of the matrix $\mathcal{B}_\alpha(\varphi_m^{(\beta)})$ can be reduced further by two successive Schur reductions according to

$$\det \begin{bmatrix} a_{11} & a_{12} \\ a_{21} & a_{22} \end{bmatrix} = \det a_{11} \det\left(a_{22} - a_{21} a_{11}^{-1} a_{12}\right), \tag{2.25}$$

where we will make the reduction with respect to upper left $2 \times 2$ block matrix, as for the first step this part solely represents the couplings in the translationally invariant, parallel direction. The first reduction gives

$$\det \mathcal{B}_{\alpha}(\varphi_m^{(\beta)}) = \det \begin{bmatrix} \mathbf{0} & J_{\beta}^+ \\ J_{\beta}^- & \mathbf{0} \end{bmatrix} \det \begin{bmatrix} -K_{\beta}^+ & J_{\alpha} - K_{\beta}^- \\ K_{\beta}^- - J_{\alpha}^{\mathsf{T}} & K_{\beta}^+ \end{bmatrix}, \tag{2.26}$$

with the matrices $K_{\beta}^{\pm} = (J_{\beta}^+)^{-1} \pm (J_{\beta}^-)^{-1}$. As both $J_{\beta}^{\pm}$ are diagonal, the first determinant simply becomes

$$\det \begin{bmatrix} \mathbf{0} & J_{\beta}^+ \\ J_{\beta}^- & \mathbf{0} \end{bmatrix} = \prod_{\ell=1}^{L} \left( 1 + t_{\ell}^* e^{i\varphi_m^{(\beta)}} \right) \left( 1 + t_{\ell}^* e^{-i\varphi_m^{(\beta)}} \right). \tag{2.27}$$

For the second determinant in (2.26) we apply a second Schur reduction with respect to the upper left block matrix, which yields

$$\det \begin{bmatrix} -K_{\beta}^+ & J_{\alpha} - K_{\beta}^- \\ K_{\beta}^- - J_{\alpha}^{\mathsf{T}} & K_{\beta}^+ \end{bmatrix} = \det \left( -K_{\beta}^+ \right) \det \left[ K_{\beta}^+ + \left( K_{\beta}^- - J_{\alpha}^{\mathsf{T}} \right) \left( K_{\beta}^+ \right)^{-1} \left( J_{\alpha} - K_{\beta}^- \right) \right]. \tag{2.28}$$

Again, the first determinant is rather simple,

$$\det \left( -K_{\beta}^+ \right) = \prod_{\ell=1}^{L} \left( \frac{1}{1 + t_{\ell}^* e^{-i\varphi_m^{(\beta)}}} - \frac{1}{1 + t_{\ell}^* e^{i\varphi_m^{(\beta)}}} \right), \tag{2.29}$$

and can be combined with (2.27) to

$$\det \left( -K_{\beta}^+ \right) \det \begin{bmatrix} \mathbf{0} & J_{\beta}^+ \\ J_{\beta}^- & \mathbf{0} \end{bmatrix} = \prod_{\ell=1}^{L} 2i t_{\ell}^* \sin \varphi_m^{(\beta)}. \tag{2.30}$$

We finally find

$$\det \mathcal{B}_{\alpha}(\varphi_m^{(\beta)}) = \det \tilde{\mathcal{C}}_{\alpha}(\varphi_m^{(\beta)}) \prod_{\ell=1}^{L} 2t_{\ell}^*, \tag{2.31}$$

with

$$\tilde{\mathcal{C}}_{\alpha}(\varphi_m^{(\beta)}) = i \sin \varphi_m^{(\beta)} \left[ K_{\beta}^+ + \left( K_{\beta}^- - J_{\alpha}^{\mathsf{T}} \right) \left( K_{\beta}^+ \right)^{-1} \left( J_{\alpha} - K_{\beta}^- \right) \right]. \tag{2.32}$$

The $L \times L$ matrix $\tilde{\mathcal{C}}_{\alpha}$ is a symmetric tridiagonal (quasi-)cyclic matrix of the form

$$\tilde{\mathcal{C}}_{\alpha}(\varphi_m^{(\beta)}) = \begin{pmatrix} a_1 & b_1 & & & b_L \\ b_1 & a_2 & \ddots & & \\ & \ddots & \ddots & b_{L-1} & \\ b_L & & b_{L-1} & a_L \end{pmatrix}, \tag{2.33a}$$

with matrix elements

$$a_{\ell > 1} = z_{\ell-1}^2 \mu_{\ell-1}^- - \mu_{\ell}^+, \tag{2.33b}$$

$$b_{\ell < L} = \frac{z_{\ell}}{(t_{\ell})_-}, \tag{2.33c}$$

and the special cases concerning whether the couplings in the perpendicular direction are periodic, anti-periodic, or open,

$$a_1 = \alpha^2 z_L^2 \mu_L^- - \mu_1^+, \tag{2.33d}$$

$$b_L = -\frac{\alpha z_L}{(t_L)_-}. \tag{2.33e}$$

Here we have used the convenient abbreviation [18]

$$x_\pm \equiv \frac{x \pm x^{-1}}{2}, \tag{2.34}$$

for an arbitrary quantity $x$, as well as

$$\mu_\ell^\pm(\varphi_m^{(\beta)}) = \cos \varphi_m^{(\beta)} \pm \frac{(t_\ell)_+}{(t_\ell)_-}, \tag{2.35}$$

which solely covers the $\varphi_m^{(\beta)}$-dependency.

The matrix $\tilde{\mathcal{C}}_\alpha$ in (2.33) will be our starting point for the different BCs, but as we are hardly interested in the case of every line having a different coupling, we will apply the changes due to the corresponding boundary conditions and then frankly assume homogenous but still anisotropic couplings. The determinant of a matrix of the form (2.33) can be easily calculated with a $2 \times 2$ transfer matrix *ansatz*, namely

$$\det \tilde{\mathcal{C}}_\alpha = \mathrm{tr}\left[ \mathcal{T}_L \cdots \mathcal{T}_2 \mathcal{T}_1 \right] - 2(-1)^L \prod_{\ell=1}^{L} b_\ell, \tag{2.36}$$

while for $b_L = 0$ the formula simplifies to

$$\det \tilde{\mathcal{C}}_\alpha = \langle 1, 0 | \mathcal{T}_L \cdots \mathcal{T}_2 \mathcal{T}_1 | 1, 0 \rangle, \tag{2.37}$$

with the transfer matrices ($b_0 \equiv b_L$)

$$\mathcal{T}_\ell = \begin{pmatrix} a_\ell & -b_{\ell-1}^2 \\ 1 & 0 \end{pmatrix}, \tag{2.38}$$

analogous to [14] for a system with open boundaries in one direction. Note that this formula is formally equivalent to the more general form in [18], where no translational invariance in the direction perpendicular to the open boundaries is assumed. Luckily the $b_\ell$ are all negative, giving an additional factor $(-1)^L$ to (2.36) and thus eliminating the dependency on whether $L$ is even or odd. Thus our results are correct for arbitrary integer $L$ and $M$.

If we assume translational invariance in both directions, we may reduce the coupling matrices further to $\boldsymbol{z}^\perp = z^\perp \mathbf{1}$ and $\boldsymbol{z}^\parallel = z^\parallel \mathbf{1}$ and diagonalise the perpendicular direction, too. Then the determinant becomes

$$\det \mathcal{A}_{\alpha\beta} = \prod_{m=0}^{M-1} \prod_{\ell=0}^{L-1} \det \begin{pmatrix} 0 & 1+z^\parallel e^{i\varphi_m^{(\beta)}} & -1 & -1 \\ -1-z^\parallel e^{-i\varphi_m^{(\beta)}} & 0 & 1 & -1 \\ 1 & -1 & 0 & 1+z^\perp e^{i\varphi_\ell^{(\alpha)}} \\ 1 & 1 & -1-z^\perp e^{-i\varphi_\ell^{(\alpha)}} & 0 \end{pmatrix} \tag{2.39a}$$

$$= \prod_{m=0}^{M-1} \prod_{\ell=0}^{L-1} 4z^\perp z^\parallel \left( z_+^\perp z_+^\parallel + z_-^\perp \cos \varphi_m^{(\alpha)} + z_-^\parallel \cos \varphi_\ell^{(\beta)} \right), \tag{2.39b}$$

which is just the result by McCoy & Wu for the anisotropic torus [17, Eq. (V.2.22)].

# 3   Scaling theory

The two-dimensional Ising model is one of the simplest systems showing a temperature-driven continuous phase transitions in the absence of a bulk magnetic field. Albeit this work focuses on the two-dimensional case, the following statements can be generalised to $d > 2$. We assume an $L \times M$ rectangular system, where $L$ is its extent in the *perpendicular* ($\perp$) direction and $M$ the extent in *parallel* ($\parallel$) direction, for reasons that will be clear later on. In the vicinity of the phase transition the thermal fluctuations of the medium become long-ranged, but by imposing some sort of BCs, even (anti-)periodicity, they are confined by the geometry. The reduced free energy[2] for arbitrary BCs $\alpha$ in perpendicular direction and translationally invariant BCs $\beta \in \{p, a\}$ in parallel direction, $F^{(\alpha,\beta)} = -\ln Z^{(\alpha,\beta)}$ (again in units of $k_B T$), may be decomposed into an infinite volume term and a residual contribution [19]

$$F^{(\alpha,\beta)}(L,M) = F_\infty^{(\alpha)}(L,M) + F_{\infty,\text{res}}^{(\alpha,\beta)}(L,M), \tag{3.1}$$

where the first term

$$F_\infty^{(\alpha)}(L,M) = LMf_b + Mf_s^{(\alpha)}, \tag{3.2}$$

with bulk and surface free energy densities $f_b$ and $f_s^{(\alpha)}$, describes the leading behaviour in the thermodynamic limit $L, M \to \infty$, while the latter one covers the finite-size effects.

Another way to decompose the free energy [18, 19] is into contributions $F_b^{(\beta)}$ for the bulk, $F_s^{(\alpha,\beta)}$ for the two surfaces, and $F_{\text{st,res}}^{(\alpha,\beta)}$ for the respective strip finite-size part as

$$F^{(\alpha,\beta)}(L,M) = \underbrace{LF_b^{(\beta)}(M) + F_s^{(\alpha,\beta)}(M)}_{F_{\text{st}}^{(\alpha,\beta)}(L,M)} + F_{\text{st,res}}^{(\alpha,\beta)}(L,M). \tag{3.3}$$

Here, $F_{\text{st}}^{(\alpha,\beta)}(L,M)$ describes the strip limit $L \to \infty$ with $M$ fixed. Combining those two decompositions splits the contributions further, namely the bulk free energy per slice $F_b^{(\beta)}$ reads

$$F_b^{(\beta)}(M) = Mf_b + F_{b,\text{res}}^{(\beta)}(M), \tag{3.4}$$

with residual part $F_{b,\text{res}}^{(\beta)}$. Analogously, we find for the surface

$$F_s^{(\alpha,\beta)}(M) = Mf_s^{(\alpha)} + F_{s,\text{res}}^{(\alpha,\beta)}(M), \tag{3.5}$$

with the corresponding residual surface free energy $F_{s,\text{res}}^{(\alpha,\beta)}(M)$. Both terms depend explicitly on the BC $\alpha$ (e.g., open boundaries or surface fields), while the residual contribution also accounts for the BC $\beta$, as these define the discrete spectrum for the finite system. Thus we can rewrite the residual contribution in (3.1) as

$$F_{\infty,\text{res}}^{(\alpha,\beta)}(L,M) = LF_{b,\text{res}}^{(\beta)}(M) + F_{s,\text{res}}^{(\alpha,\beta)}(M) + F_{\text{st,res}}^{(\alpha,\beta)}(L,M), \tag{3.6}$$

cf. [19]. Note that it is possible to impose boundaries in more than one direction, introducing edges and corners and suitable contributions depending on the dimension of the hypercuboid; for the two dimensional case of the Ising model on the open rectangle the necessary calculations were done lately with the dimer approach presented above [18, 19], while the corresponding leading term $F_\infty(L,M)$ was analysed in detail within the spinor representation by Baxter [9].

---

[2]In the following we will omit the explicit dependency on the temperature $T$.

**General scaling behaviour**

Near criticality the bulk correlation length in direction $\delta$ diverges as

$$\xi_\delta^{(\infty)}(\tau) \overset{\tau > 0}{\simeq} \hat{\xi}_\delta \tau^{-\nu}, \tag{3.7}$$

where $\tau = T/T_c - 1$ is the reduced temperature and $\hat{\xi}_\delta$ is the correlation length amplitude in the unordered phase, while $\nu$ is the associated scaling exponent, with $\nu = 1$ for the two-dimensional case. In the region around $T_c$ where the correlation length is the dominant length scale, the residual free energy $F_{\infty,\text{res}}^{(\alpha,\beta)}$ only depends on the two length ratios $L/\xi_\perp^{(\infty)}(\tau)$ and $M/\xi_\parallel^{(\infty)}(\tau)$, where we assume the bulk correlation length amplitudes differ depending on the direction because of the anisotropic couplings. Following Fisher & de Gennes [29], the residual free energy thus fulfils the scaling *ansatz*

$$F_{\infty,\text{res}}^{(\alpha,\beta)}(L,M) \simeq \Theta^{(\alpha,\beta)}(x_\perp, x_\parallel), \tag{3.8}$$

where $\Theta^{(\alpha,\beta)}$ is the total residual free energy scaling function, depending on the two temperature scaling variables

$$x_\perp \equiv \tau \left( \frac{L}{\hat{\xi}_\perp} \right)^{1/\nu}, \qquad\qquad x_\parallel \equiv \tau \left( \frac{M}{\hat{\xi}_\parallel} \right)^{1/\nu}, \tag{3.9}$$

which are related by the reduced aspect ratio

$$\rho \equiv \frac{L/\hat{\xi}_\perp}{M/\hat{\xi}_\parallel} \tag{3.10}$$

via the relation

$$x_\perp = \rho^{1/\nu} x_\parallel. \tag{3.11}$$

Additionally, it is sometimes [68] advantageous to consider scaling functions depending on a volume-like scaling variable

$$x_\circ \equiv \tau \left( \frac{LM}{\hat{\xi}_\perp \hat{\xi}_\parallel} \right)^{\frac{1}{2\nu}}. \tag{3.12}$$

Consequently, we may change our focus onto either one of the two directions or the volume and rewrite the scaling functions accordingly into a perpendicular ($\perp$), a parallel ($\parallel$), or a volume ($\circ$) form

$$F_{\infty,\text{res}}^{(\alpha,\beta)}(L,M) \simeq \Theta_\circ^{(\alpha,\beta)}(x_\circ, \rho) = \rho^{-1} \Theta_\perp^{(\alpha,\beta)}(x_\perp, \rho) = \rho\, \Theta_\parallel^{(\alpha,\beta)}(x_\parallel, \rho). \tag{3.13}$$

The critical Casimir force is defined as derivative of the residual free energy with respect to the corresponding system length, e. g. in perpendicular direction,

$$\mathcal{F}_C^{(\alpha,\beta)}(L,M) \equiv -\frac{1}{M} \frac{\partial}{\partial L} F_{\infty,\text{res}}^{(\alpha,\beta)}(L,M) \tag{3.14}$$

and analogously to (3.13) scales as

$$\mathcal{F}_C^{(\alpha,\beta)}(L,M) \simeq (LM)^{-1} \vartheta_\circ^{(\alpha,\beta)}(x_\circ, \rho) = L^{-2} \vartheta_\perp^{(\alpha,\beta)}(x_\perp, \rho) = M^{-2} \vartheta_\parallel^{(\alpha,\beta)}(x_\parallel, \rho). \tag{3.15}$$

The residual free energy and the Casimir force scaling functions are connected by the relations [68, 79]

$$\vartheta_\perp^{(\alpha,\beta)}(x_\perp,\rho) = -\left[-1 + \frac{x_\perp}{\nu}\frac{\partial}{\partial x_\perp} + \rho\frac{\partial}{\partial \rho}\right]\Theta_\perp^{(\alpha,\beta)}(x_\perp,\rho), \tag{3.16a}$$

$$\vartheta_\parallel^{(\alpha,\beta)}(x_\parallel,\rho) = -\left[1 + \rho\frac{\partial}{\partial \rho}\right]\Theta_\parallel^{(\alpha,\beta)}(x_\parallel,\rho), \tag{3.16b}$$

$$\vartheta_\circ^{(\alpha,\beta)}(x_\circ,\rho) = -\left[\frac{x_\circ}{2\nu}\frac{\partial}{\partial x_\circ} + \rho\frac{\partial}{\partial \rho}\right]\Theta_\circ^{(\alpha,\beta)}(x_\circ,\rho). \tag{3.16c}$$

In the following we will focus on the scaling function $\Theta_\parallel^{(\alpha,\beta)}(x_\parallel,\rho)$ for the parallel direction, as it arises naturally within our calculations from the product in (2.22). Just like the residual free energy can be decomposed into its bulk, surface, and strip contribution in (3.6), the scaling function consists of such parts [19], which can be written as

$$\rho\,\Theta_\parallel^{(\alpha,\beta)}(x_\parallel,\rho) = \rho\,\Theta_b^{(\beta)}(x_\parallel) + \Theta_s^{(\alpha,\beta)}(x_\parallel) + \Psi^{(\alpha,\beta)}(x_\parallel,\rho), \tag{3.17}$$

with

$$LF_{b,res}^{(\beta)}(M) \simeq \rho\,\Theta_b^{(\beta)}(x_\parallel), \tag{3.18a}$$

$$F_{s,res}^{(\alpha,\beta)}(M) \simeq \Theta_s^{(\alpha,\beta)}(x_\parallel), \tag{3.18b}$$

$$F_{st,res}^{(\alpha,\beta)}(L,M) \simeq \Psi^{(\alpha,\beta)}(x_\parallel,\rho). \tag{3.18c}$$

For the torus and the other systems with translational invariance in both directions there is no surface and thus $\Theta_s^{(\alpha,\beta)}(x_\parallel) \equiv 0$ for $\alpha,\beta \in \{p,a\}$, all other cases will be discussed in the second part of this paper [28].

**Anisotropic scaling**

Now we need to discuss the influence of the *weakly anisotropic* couplings on the scaling behaviour. Here the critical point expands to a critical line as we may rewrite the condition of criticality [80] as

$$\sinh(2K^\perp)\sinh(2K^\parallel) = 1 \quad\Leftrightarrow\quad t = z, \tag{3.19}$$

which again becomes a point by fixing the couplings by a factor $\kappa$ as

$$K^\parallel = \kappa K^\perp. \tag{3.20}$$

In terms of $t$ and $z$ this becomes

$$t = (z^*)^\kappa, \tag{3.21}$$

and we can identify the critical point with the equation

$$z_c(\kappa) = \left[\frac{1 - z_c(\kappa)}{1 + z_c(\kappa)}\right]^\kappa. \tag{3.22}$$

The correlation lengths along the two directions read [17]

$$\xi_\perp^{(\infty)}(z,t) = \left(\ln\coth K^\perp - 2K^\parallel\right)^{-1} = \ln^{-1}\frac{t}{z}, \tag{3.23a}$$

$$\xi_\parallel^{(\infty)}(z,t) = \left(\ln\coth K^\parallel - 2K^\perp\right)^{-1} = \ln^{-1}\frac{z^*}{t^*}, \tag{3.23b}$$

with the dual couplings $z^*$ and $t^*$ from (2.24). This form emphasises the choice of the reduced temperatures in the two directions as

$$\tau_\perp = \frac{t}{z} - 1, \qquad\qquad \tau_\parallel = \frac{z^*}{t^*} - 1. \qquad (3.24)$$

The amplitude ratio $r_\xi \equiv \hat{\xi}_\perp/\hat{\xi}_\parallel \simeq \xi_\perp^{(\infty)}/\xi_\parallel^{(\infty)}$ from [81] can be expanded around criticality $t = z$ using $\log(1+x) \simeq x$ to give $r_\xi \simeq \tau_\parallel/\tau_\perp$, such that the scaling variables fulfil

$$x_\perp \simeq L\tau_\perp, \qquad\qquad x_\parallel \simeq M\tau_\parallel. \qquad (3.25)$$

## 4 The torus

After we have calculated the partition function on the finite lattice, we will carve out the bulk properties and calculate its thermodynamic limit. Afterwards we will see how the dimer approach and the choice of signs distinguishes between periodicity and antiperiodic BCs as a consequence of its interplay with the directed graph being subject to the Pfaffian.

Let us return to (2.33), for which we now assume independent homogeneity in both directions, i.e., $z_\ell \equiv z \ \forall \ \ell$ and $t_\ell \equiv t \ \forall \ \ell$, as well as $\mu_\ell^\pm(\varphi) \equiv \mu^\pm(\varphi) \ \forall \ \ell$. Then it is comfortable to factorise the homogeneous factor $-z/t_-$, leaving every non-zero off-diagonal element but $b_L$ equal to $-1$. Due to this procedure, the diagonal entries simplify to

$$\frac{t_-}{z}\left[\mu^+(\varphi_m^{(\beta)}) - z^2\mu^-(\varphi_m^{(\beta)})\right] = 2\left(t_+z_+ - t_-z_- \cos\varphi_m^{(\beta)}\right), \qquad (4.1)$$

which can be parameterised into the anisotropic Onsager dispersion relation

$$\cosh\gamma_m^{(\beta)} \equiv t_+z_+ - t_-z_- \cos\varphi_m^{(\beta)}. \qquad (4.2)$$

Thus the matrix for the torus has the form

$$\mathcal{C}_L^{(\alpha)}(\varphi_m^{(\beta)}) = \begin{pmatrix} 2\cosh\gamma_m^{(\beta)} & -1 & & \alpha \\ -1 & 2\cosh\gamma_m^{(\beta)} & \ddots & \\ & \ddots & \ddots & -1 \\ \alpha & & -1 & 2\cosh\gamma_m^{(\beta)} \end{pmatrix}, \qquad (4.3)$$

which is connected to (2.33) by

$$\det\tilde{\mathcal{C}}_\alpha(\varphi_m^{(\beta)}) = \left(-\frac{z}{t_-}\right)^L \det\mathcal{C}_L^{(\alpha)}(\varphi_m^{(\beta)}). \qquad (4.4)$$

To calculate this determinant, we use the transfer matrix approach (2.36) to find

$$\det\mathcal{C}_L^{(\alpha)}(\varphi_m^{(\beta)}) = \mathrm{tr}\left[\mathcal{T}^L(\gamma_m^{(\beta)})\right] + 2\alpha, \qquad (4.5)$$

with the transfer matrix

$$\mathcal{T}(\gamma_m^{(\beta)}) = \begin{pmatrix} 2\cosh\gamma_m^{(\beta)} & -1 \\ 1 & 0 \end{pmatrix}. \qquad (4.6)$$

The eigenvalues of $\mathcal{T}$ are $\mathrm{e}^{\pm\gamma_m^{(\beta)}}$ and, depending on $\alpha = \pm 1$, the determinant thus reads

$$\det\mathcal{C}_L^{(\alpha)}(\varphi_m^{(\beta)}) = \left(\mathrm{e}^{\frac{L}{2}\gamma_m^{(\beta)}} + \alpha\,\mathrm{e}^{-\frac{L}{2}\gamma_m^{(\beta)}}\right)^2. \qquad (4.7)$$

In the following we will omit the superscript of the $\gamma$, since the parity is encoded in the product over either even or odd numbers. Having said that, we denote the parity in the other direction with $\pm$ to account for the dependency on $\alpha$, and thus we use

$$Z_{e/o}^{\pm} = \prod_{\substack{0 \le m < 2M \\ m \, \text{even/odd}}} \left( e^{\frac{L}{2}\gamma_m} \pm e^{-\frac{L}{2}\gamma_m} \right), \tag{4.8}$$

where we have incorporated the square root of (2.5). Then the non-regular part of the partition function reads

$$\frac{Z^{(\text{p,p})}}{Z_0^{(\text{p,p})}} = \frac{1}{2} \left( \frac{2z}{1+t_+} \right)^{\frac{LM}{2}} \left[ Z_o^+ + Z_o^- + Z_e^+ - \text{sgn}(t-z) Z_e^- \right], \tag{4.9}$$

with $Z_0^{(\text{p,p})}$ from (2.3), where the sgn$(t-z)$ stems from the root in (2.5), too, and assures that for arbitrary anisotropy $\kappa$ the contribution is a monotonic function in temperature. This can be easily understood by looking at the critical value of $Z_e^-$: There $\gamma_0 = 0$ and thus one of the factors becomes zero, but as we have taken the square root in (4.9), we have to correct the respective sign. Of course this is just exactly the result by Kaufman [2].

## Bulk and finite-size contribution

In section 3 we discussed how the free energy of a finite system may be decomposed into summands describing the different contributions from volume, surfaces, and finiteness. As a matter of fact, the toroidal geometry has no surfaces and thus the aforementioned decomposition includes only the bulk contribution and the finite-size part. Therefore it is perfectly suitable to identify the former one and calculate its thermodynamic limit.

We follow the procedure of Ferdinand & Fisher [74] and start by splitting the $Z_{e/o}^{\pm}$ into their exponentially growing and decaying parts as

$$Z_{e/o}^{\pm}(L,M) = p_{e/o}^{\pm}(L,M) \prod_{\substack{0 \le m < 2M \\ m \, \text{even/odd}}} e^{\frac{L}{2}\gamma_m}, \tag{4.10}$$

with

$$p_{e/o}^{\pm}(L,M) = \prod_{\substack{0 \le m < 2M \\ m \, \text{even/odd}}} \left( 1 \pm e^{-L\gamma_m} \right). \tag{4.11}$$

Now we may factorise the odd product over the leading exponential in (4.9) from *all four* contributions, as it is slightly larger than the one over the even numbers. The corresponding bulk part of the free energy then simply reads

$$F_b^{(\text{p})}(M) = -\frac{M}{2} \left[ \ln \frac{2z}{1+t_+} + \frac{1}{M} \sum_{\substack{0 \le m < 2M \\ m \, \text{odd}}} \gamma_m \right], \tag{4.12}$$

thus leaving the residual part as

$$F_{\text{st,res}}^{(\text{p,p})}(L,M) = -\ln \left[ \frac{p_o^+ + p_o^-}{2} + \frac{p_e^+ - \text{sgn}(t-z) p_e^-}{2} \prod_{m=0}^{2M-1} e^{(-1)^m \frac{L}{2}\gamma_m} \right]. \tag{4.13}$$

To calculate the bulk contribution to the free energy in the thermodynamic limit,

$$f_b(t,z) \equiv \lim_{M \to \infty} M^{-1} F_b^{(\text{p})}(M), \tag{4.14}$$

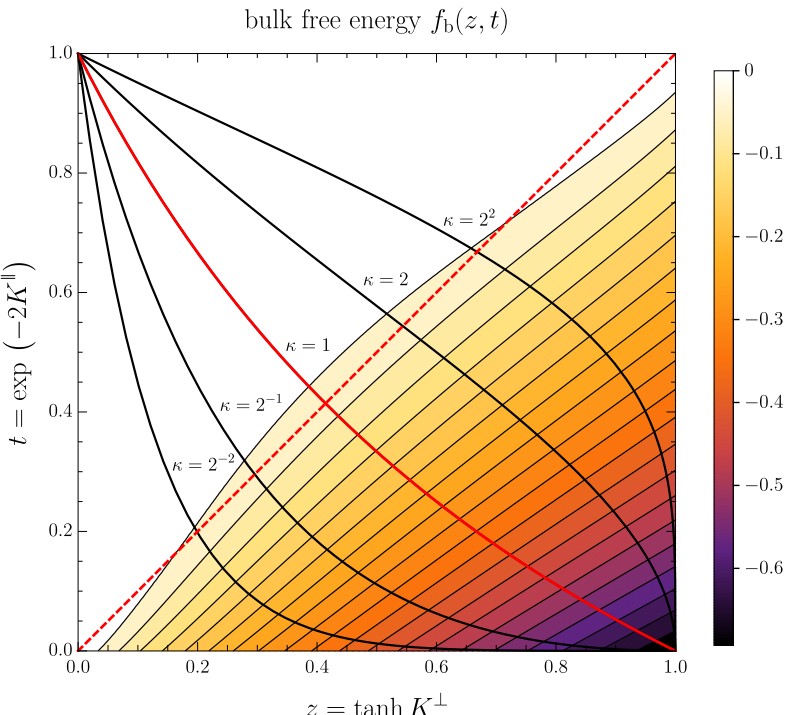

Figure 4: Bulk free energy density $f_b(z, t)$. The dashed line marks criticality for arbitrary anisotropy $\kappa = K^\perp/K^\parallel$. For fixed anisotropy the two couplings are connected by $t = (z^*)^\kappa$, see (3.21), and the black lines mark the run of the corresponding curve, where the isotropic case $\kappa = 1$ is shown in red.

we use the Euler-Maclaurin sum formula to obtain the integral representation

$$f_b(z, t) = -\frac{1}{2} \ln \frac{2z}{1 + t_+} - \frac{1}{4\pi} \int_0^{2\pi} d\varphi \, \gamma(\varphi) \tag{4.15}$$

and omit all corrections of $\mathcal{O}(M^{-1})$ or higher. This is in perfect agreement with former results by McCoy & Wu [14]. Fig. 4 shows the bulk free energy density as function of the two coupling variables $z$ and $t$. For the isotropic critical case we find

$$f_b(z_c, z_c) = -\ln(1 - z_c) - \frac{2G}{\pi}, \tag{4.16}$$

where $G$ is Catalan's constant. Together with the contribution from the regular part of the partition function, this coincides perfectly with the result by Izmailian [60].

**Scaling form**

As discussed in Sec. 3, in a finite system there is always not only a residual contribution present, but this contribution can be split up into different parts as well. We start with the scaling form of $\gamma$ as it is the central quantity for the toroidal geometry, afterwards we will first calculate the residual bulk free energy scaling function and then the one for the residual strip free energy. Therefore we introduce the hyperbolic parametrisation based on the scaling form of $\gamma$, which allows us to regularise the arising terms properly. At criticality the bulk residual scaling function coincides with the change of the free energy of the associated conformal field theory due to the projection from the plane onto the cylinder.

For the scaling limit of $\gamma$ we start with the definition for the anisotropic temperature scaling variables in (3.24) and (3.25) and solve them for $z$ and $t$ to get the somehow unhandy formulas

$$t(r_\xi, M; x_\parallel) = \frac{1}{r_\xi}\left(1 + \frac{x_\parallel}{2M}\right)\left[\sqrt{1 + r_\xi^2\frac{1 + \frac{x_\parallel}{r_\xi M}}{\left(1 + \frac{x_\parallel}{2M}\right)^2}} - 1\right], \tag{4.17a}$$

$$z(r_\xi, M; x_\parallel) = \frac{t(r_\xi, M; x_\parallel)}{1 + \frac{x_\parallel}{r_\xi M}}. \tag{4.17b}$$

Now we use them in (4.2) together with $\varphi = \Phi/M$, substitute $M = \epsilon^{-1}$ and expand $\cos(\epsilon\Phi)$ up to the second order in $\epsilon$ around $\epsilon = 0$ to obtain

$$\cosh\gamma = 1 + \frac{\epsilon^2}{2}\frac{\Gamma^2 + \epsilon x_\parallel\Phi^2}{r_\xi^2 + \epsilon r_\xi x_\parallel} + \mathcal{O}(\epsilon^4), \tag{4.18}$$

where we have introduced

$$\Gamma = \sqrt{x_\parallel^2 + \Phi^2}. \tag{4.19}$$

With a Puiseux series expansion around $\epsilon = 0$,

$$\text{arcosh}\left(1 + \frac{y^2}{2}\right) = y + \mathcal{O}\left(y^3\right), \tag{4.20}$$

we finally find, with $L = \rho\, r_\xi M$,

$$L\gamma(\varphi) \simeq \rho\lim_{\epsilon\to 0}\left(\frac{\Gamma^2 + \epsilon\, x_\parallel\Phi^2}{1 + \epsilon\frac{x_\parallel}{r_\xi}}\right)^{1/2} = \rho\,\Gamma. \tag{4.21}$$

As we made only the most general assumptions, we can use this result throughout the whole rest of this work. We finally note that (4.19), rewritten as

$$\Gamma^2 = x_\parallel^2 + \Phi^2 \tag{4.22}$$

is the finite-size scaling form of the Onsager dispersion relation (4.2) just as in the isotropic case [19].

Now we will turn to the characteristic polynomials (2.21). Their scaling form is quite simple as we only have to replace $\varphi = \Phi/M$ to obtain

$$\mathcal{P}_e^\pm(\Phi) = e^{\pm i\Phi} - 1, \tag{4.23a}$$

$$\mathcal{P}_o^\pm(\Phi) = e^{\pm i\Phi} + 1, \tag{4.23b}$$

where the $\pm$ accounts for the two possible choices of the eigenvalue $e^{\pm i\varphi_m^{(\beta)}}$ in (2.19). This freedom is essential to the regularisation of the residual free energies, because if we assume $\Phi$ to be complex to calculate the sum in terms of contour integrals, $\mathcal{P}_{e/o}^+(\Phi)$ diverges in the lower and $\mathcal{P}_{e/o}^-(\Phi)$ diverges in the upper half-plane. To avoid these divergences we can switch between the two possible realisations if we cross the real axis. We obtain suitable counting polynomials for the even and odd sums as integration kernel from the logarithmic derivative of the $\mathcal{P}_{e/o}^\pm(\Phi)$ as

$$\mathcal{K}_e^\pm(\Phi) = \frac{\partial}{\partial\Phi}\ln\mathcal{P}_e^\pm(\Phi) = +\frac{1}{2}\left[\cot\frac{\Phi}{2}\pm i\right], \tag{4.24a}$$

$$\mathcal{K}_o^\pm(\Phi) = \frac{\partial}{\partial\Phi}\ln\mathcal{P}_o^\pm(\Phi) = -\frac{1}{2}\left[\tan\frac{\Phi}{2}\mp i\right], \tag{4.24b}$$

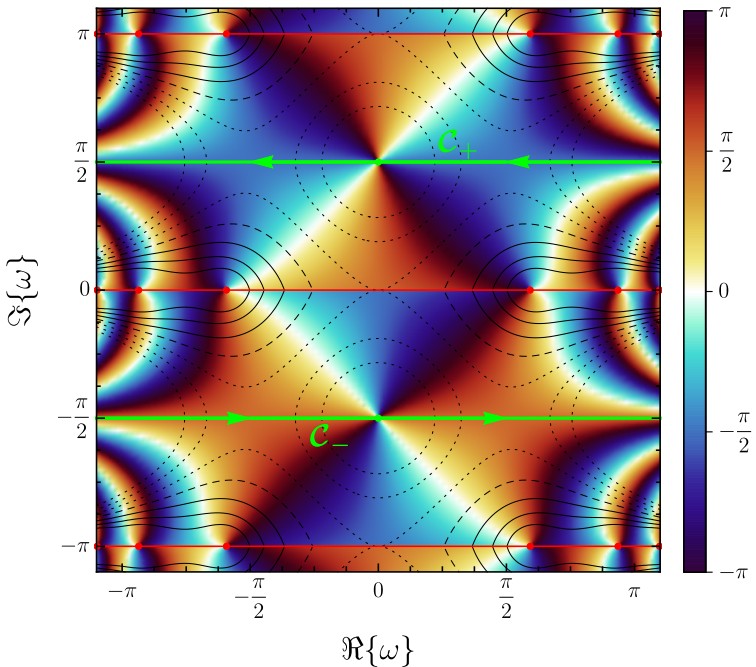

Figure 5: Complex structure of the integrand of (4.27). The two contours $\mathcal{C}_+$ in the upper and $\mathcal{C}_-$ in the lower half-plane are connected by non-contributing paths at $\Re\{\omega\} \to \pm\infty$ to form the closed contour $\mathcal{C}$. The complex phase is colour coded from $-\pi$ to $\pi$, while the lines of constant absolute value $c$ are shown as black dotted ($c < 1$), dashed ($c = 1$) or solid ($c > 1$) lines ranging from $2^{-3}$ to $2^{+3}$. The zeros at $\omega = \pm i\pi/2$ are marked as green dots, while the poles on the lines with $\Im\{\omega\} = n\pi$, $n \in \mathbb{Z}$, are marked as red dots. Additionally, the phase jump due to the different choices for the counting polynomial for the upper and lower half-plane are marked as red lines. Note that due to the transformation into the hyperbolic $\omega$-plane the structure is $2\pi$-periodic along the imaginary axis. The contributions along the paths $\mathcal{C}_+$ and $\mathcal{C}_-$ are equal, as the phase in the upper half-period is reversed with respect to the lower half-period.

and consequently for the alternating sum

$$\delta\mathcal{K}(\Phi) = \frac{\partial}{\partial\Phi} \ln \frac{\mathcal{P}_{\mathrm{e}}^{\pm}(\Phi)}{\mathcal{P}_{\mathrm{o}}^{\pm}(\Phi)} = \csc\Phi, \qquad (4.24c)$$

without any distinction for the upper and lower half-plane, as the divergences cancel each other.

Looking closely at (4.19) we see that we can parametrise this hyperbolic equation by

$$\Phi = |x_{\parallel}| \sinh\omega, \qquad (4.25a)$$

$$\Gamma = |x_{\parallel}| \cosh\omega, \qquad (4.25b)$$

solving the equation for arbitrary $\omega \in \mathbb{C}$. Note that this is the critical limiting case of the elliptic parametrisation of (4.2), which is used off criticality for finite systems with more complex BCs, e. g., for the open rectangle [9].

Combining (4.21), (4.24b) and (4.25) we now have the tools to calculate the scaling func-

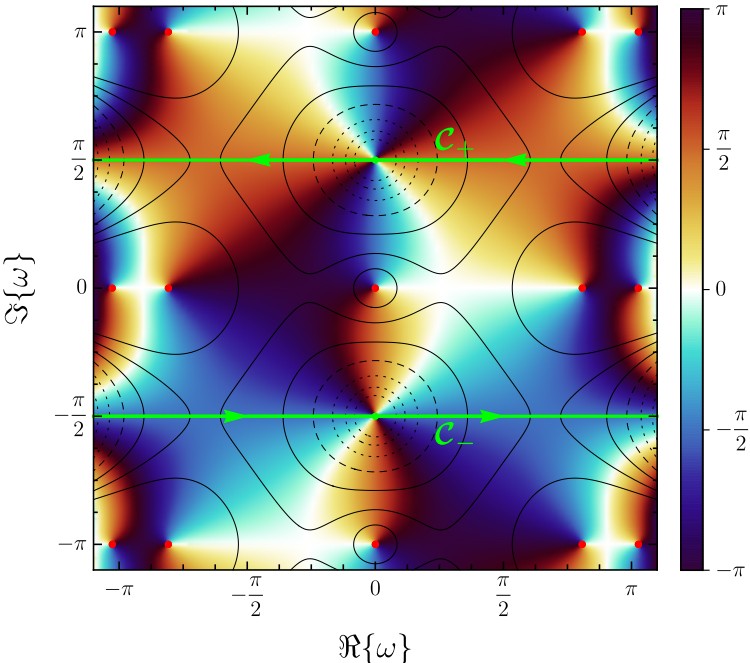

Figure 6: Complex structure of the alternating sum over the $\Gamma_m$ in the hyperbolic $\omega$-plane (integrand of (4.31a)), together with the contour $\mathcal{C} = \mathcal{C}_+ + \mathcal{C}_-$. Here no phase jump is present and the $2\pi$-periodicity along the imaginary axis is continuous, but nevertheless the upper and lower contour are equal due to the reversed phase in the upper and lower half-period.

tion of the residual bulk free energy

$$F_{\text{res,b}}^{(\text{p})}(M) = F_{\text{b}}^{(\text{p})}(M) - M f_{\text{b}} = \frac{M}{4\pi} \int_0^{2\pi} \mathrm{d}\varphi \, \gamma(\varphi) - \frac{1}{2} \sum_{\substack{0 \le m < 2M \\ m \, \text{odd}}} \gamma_m. \tag{4.26}$$

By rewriting the sum as contour integral over $\Gamma$ with the counting polynomial $\mathcal{K}_{\text{o}}^{\pm}$ and substituting the hyperbolic parametrisation we get

$$\Theta_{\text{b}}^{(\text{p})}(x_\parallel) = -\frac{1}{4\mathrm{i}\pi} \oint_{\mathcal{C}} \mathrm{d}\omega \, x_\parallel^2 \cosh^2 \omega \, \mathcal{K}_{\text{o}}^{\pm} \left( |x_\parallel| \sinh \omega \right). \tag{4.27}$$

Note that the additional term $\pm \mathrm{i}$ in $\mathcal{K}_{\text{o}}^{\pm}$ can be interpreted as the integral in (4.26) for the bulk free energy, as

$$\lim_{\Phi \to \pm \mathrm{i}\infty} \tan \frac{\Phi}{2} = \pm \mathrm{i}, \tag{4.28}$$

and thus a shift of the integration path along the imaginary axis can be interpreted as smooth interpolation between the sum on the real axis and the integral at infinity, which is mapped onto the lines at $\pm \mathrm{i}\pi/2$ within the hyperbolic parametrisation. The integrand is shown in Fig. 5 together with an appropriately chosen contour. For $|\Phi| \to \infty$ the integrand vanishes, thus making the integral over the paths $\Im\{\omega\} = \pm \pi/2$ the only relevant parts of the contour,

which are equal as they only differ by a phase of $\pi$. This leaves us with

$$\Theta_{\mathrm{b}}^{(\mathrm{p})}(x_{\|}) = \frac{1}{2\mathrm{i}\pi} \int_{-\infty}^{\infty} \mathrm{d}\omega\, x_{\|}^2 \sinh^2\omega\, \frac{\mathrm{i}}{2}\left[ \tanh\left( \frac{|x_{\|}|}{2}\cosh\omega \right) - 1 \right] \tag{4.29a}$$

$$= \frac{1}{4\pi} \int_{-\infty}^{\infty} \mathrm{d}\Phi\, \frac{\Phi^2}{\Gamma}\left[ \tanh\frac{\Gamma}{2} - 1 \right] \tag{4.29b}$$

$$= -\frac{1}{2\pi} \int_{-\infty}^{\infty} \mathrm{d}\Phi\, \ln\left[ 1 + \mathrm{e}^{-\Gamma} \right], \tag{4.29c}$$

where we first resubstituted to the $\Phi$-plane and then integrated by parts, reproducing the result from [73] calculated for the isotropic case.

A similar procedure will be our way to go in every upcoming calculation, that is, use the associated counting polynomial as integral kernel in the contour integral representation of the sum, transform it into the hyperbolic parametrisation, shift the integration path to $\mathfrak{I}\{\omega\} = \pm\pi/2$, and perform the integration. Applying this scheme to the alternating sum in the exponent of (4.13),

$$\frac{L}{2} \sum_{m=0}^{2M-1} (-1)^m \gamma_m \simeq \rho\, \delta\Theta_{\mathrm{b}}(x_{\|}), \tag{4.30}$$

then gives us

$$\delta\Theta_{\mathrm{b}}(x_{\|}) = \frac{1}{4\mathrm{i}\pi} \int_{\mathcal{C}} \mathrm{d}\omega\, x_{\|}^2 \cosh^2\omega\, \delta\mathcal{K}(|x_{\|}|\sinh\omega) \tag{4.31a}$$

$$= \frac{1}{2\pi} \int_{-\infty}^{\infty} \mathrm{d}\Phi\, \frac{\Phi^2}{\Gamma}\operatorname{csch}\Gamma \tag{4.31b}$$

$$= -\frac{1}{2\pi} \int_{-\infty}^{\infty} \mathrm{d}\Phi\, \ln\frac{1 - \mathrm{e}^{-\Gamma}}{1 + \mathrm{e}^{-\Gamma}}, \tag{4.31c}$$

again reproducing the result of [73]. Note that the alternating sum regularises itself and thus there is only the well known kernel for alternating sums present, which is shown in Fig. 6 in the same manner as Fig. 5.

Lastly we have to calculate the scaling limit of the $p_{\mathrm{e/o}}^{\pm}(L, M)$, which is really straight forward, as we only have to expand the product symmetrically to infinity and replace $L\gamma$ by its scaling form. Thus we get

$$p_{\mathrm{e/o}}^{\pm}(L, M) \simeq P_{\mathrm{e/o}}^{\pm}(x_{\|}, \rho), \tag{4.32}$$

with

$$P_{\mathrm{e/o}}^{\pm}(x_{\|}, \rho) = \prod_{\substack{m=-\infty \\ m\,\mathrm{even/odd}}}^{\infty} \left( 1 \pm \mathrm{e}^{-\rho\,\Gamma_m} \right), \tag{4.33}$$

where we have used the discrete form of $\Gamma$,

$$\Gamma_m = \sqrt{x_{\|}^2 + \Phi_m^2}, \tag{4.34}$$

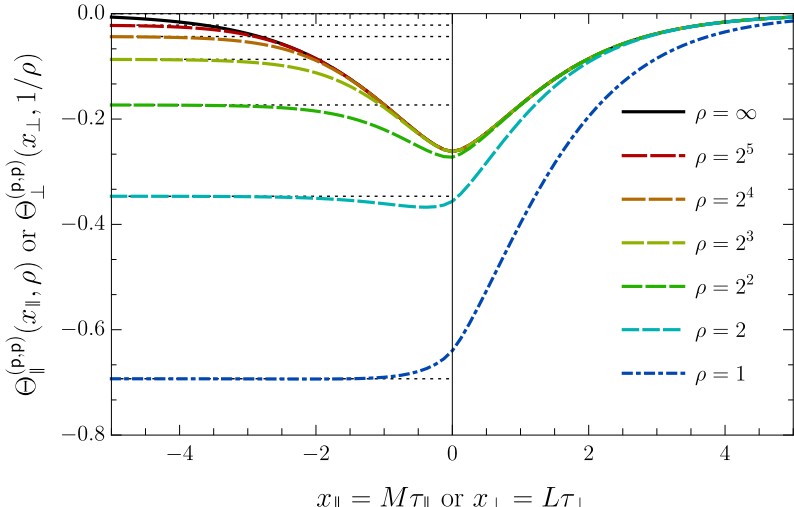

Figure 7: Scaling function $\Theta_\delta^{(p,p)}(x_\delta, \rho)$ for different values for the aspect ratio $\rho$. Due to the periodic BCs the system has the symmetry $\Theta_\parallel^{(p,p)}(x_\parallel, \rho) = \Theta_\perp^{(p,p)}(x_\perp, 1/\rho)$. The limit (4.39) for $x_\delta \to -\infty$ due to the degeneracy is shown for corresponding values of $\rho$ as dotted lines.

with $\Phi_m = m\pi$, and used the symmetries of $\gamma_m$ with respect to $m = 0$ and $m = M$ to expand the product symmetric around zero. Now we combine them into an even and an odd scaling function

$$\Psi_e^{(p,p)}(x_\parallel, \rho) = -\ln \frac{P_e^+(x_\parallel, \rho) - \mathrm{sgn}(x_\parallel) P_e^-(x_\parallel, \rho)}{2}, \tag{4.35a}$$

$$\Psi_o^{(p,p)}(x_\parallel, \rho) = -\ln \frac{P_o^+(x_\parallel, \rho) + P_o^-(x_\parallel, \rho)}{2}, \tag{4.35b}$$

and calculate the analogue of an alternating strip contribution

$$\delta\Psi^{(p,p)}(x_\parallel, \rho) = \Psi_e^{(p,p)}(x_\parallel, \rho) - \Psi_o^{(p,p)}(x_\parallel, \rho). \tag{4.36}$$

Finally we find the scaling function for the torus to be

$$\Theta_\parallel^{(p,p)}(x_\parallel, \rho) = \Theta_b^{(p)}(x_\parallel) + \rho^{-1}\Psi_o^{(p,p)}(x_\parallel, \rho) - \rho^{-1}\ln\left[1 + e^{-\rho\delta\Theta_b(x_\parallel) - \delta\Psi^{(p,p)}(x_\parallel, \rho)}\right], \tag{4.37}$$

see Fig. 7, which has some remarkable properties. First and foremost this form shows the symmetry $\Theta_\parallel^{(p,p)}(x_\parallel, \rho) = \Theta_\perp^{(p,p)}(x_\perp, 1/\rho)$, as $\Theta_b^{(p)}$ and $\Psi_o^{(p,p)}$ as well as $\delta\Theta_b$ and $\delta\Psi^{(p,p)}$ exchange their roles under this transformations, which may be seen best in the two limiting cases

$$\Theta_\parallel^{(p,p)}(x_\parallel, \rho \to \infty) = \Theta_b^{(p)}(x_\parallel), \tag{4.38a}$$

$$\Theta_\perp^{(p,p)}(x_\perp, \rho \to 0) = \Theta_b^{(p)}(x_\perp), \tag{4.38b}$$

where for the latter one $P_o^+$ becomes dominant near $x_\perp = 0$ and equal to $P_o^-$ for $|x_\perp| \gg 1$, thus $\Psi_o^{(p,p)}$ can be written as integral in terms of the Euler-Maclaurin formula. Additionally the last term of (4.37) is only important for finite $\rho$ and even dominant in the vicinity of $\rho = 1$ with

$$\lim_{x_\parallel \to -\infty} \ln\left[1 + e^{-\rho\delta\Theta_b(x_\parallel) - \delta\Psi^{(p,p)}(x_\parallel, \rho)}\right] = \ln 2, \tag{4.39}$$

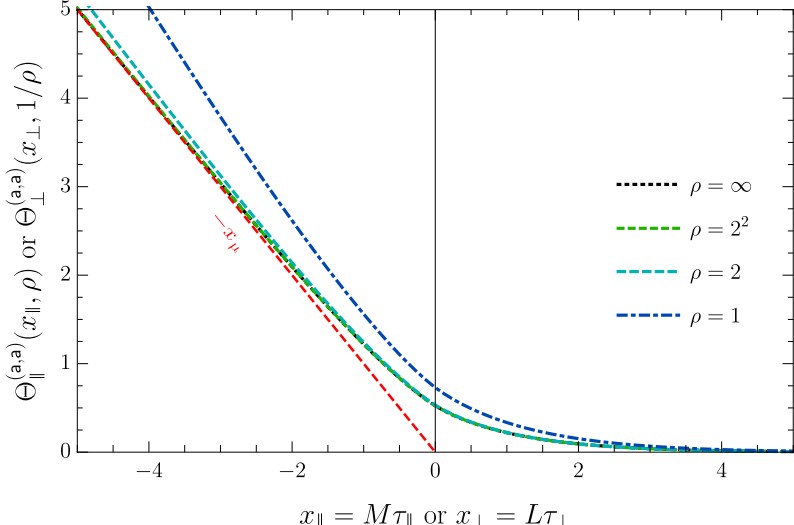

Figure 8: Scaling function $\Theta_\delta^{(a,a)}(x_\delta, \rho)$ for different values for the aspect ratio $\rho$. Due to the antiperiodic BCs the system has the symmetry $\Theta_\parallel^{(a,a)}(x_\parallel, \rho) = \Theta_\perp^{(a,a)}(x_\perp, 1/\rho)$. For negative values of $x_\delta$ the scaling function diverges linearly, marked as red (dark) dashed line. The scaling function converges very fast to its limiting form at $\rho = \infty$, while its biggest contribution is at $\rho = 1$ as there the domain wall lies diagonal in the system and is thus longest compared to the two length scales $L$ and $M$.

which stems from the different order of the limits in the scaling regime and the thermodynamic limit. While in an infinitely large system as supposed by the latter one, the system is frozen in either of the two possible magnetised states because a transition would require infinitely much energy, the finiteness of the system in the scaling limit allows such a transition, resulting in a factor 2 in the partition function and thus this topological contribution in the scaling function. However, whenever this symmetry is broken, e.g., by a surface field, this contribution is not present and the scaling functions decay to zero for $|x_\delta| \rightarrow \infty$. Indeed our results for the torus coincide with the results by Ferdinand & Fisher [74], as well as with the more recent results in [68, 73], although we did not assume isotropy. Thus we showed explicitly for finite-size scaling functions of the Ising model that a coupling anisotropy can be absorbed into a generalised aspect ratio $\rho$ as proposed, e.g., in [81], and that therefore the universal finite-size scaling functions are independent of the coupling anisotropy.

## 5 Antiperiodicity and surface tension

Let us now turn back to the finite systems solution using the dimer approach and consider antiperiodic boundary conditions in the parallel direction, i.e, $\sigma_{\ell,m} \equiv -\sigma_{\ell,m+M}$. They can be implemented as one line of anti-ferromagnetic parallel couplings, which corresponds to a change in the orientation of the oriented lattice concerning this particular line within the dimer representation. Here we need to emphasise that the oriented graph from the dimer mapping is absolutely independent of the choice of sign of any coupling. Nevertheless, changing the sign of *all* couplings in one row (or column) can be interpreted as reversing the direction of all corresponding edges and leaving the couplings unchanged. We now may choose the line, which imposes these BCs to be the line concerning the handling of the transition circles

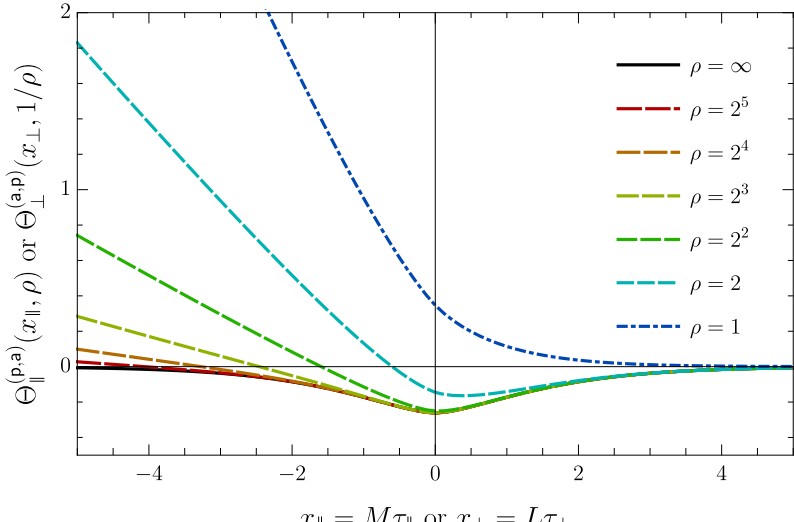

Figure 9: Scaling functions $\Theta_{\parallel}^{(p,a)}(x_{\parallel}, \rho)$ and due to the symmetry $\Theta_{\perp}^{(a,p)}(x_{\perp}, 1/\rho)$ for different values for the aspect ratio $\rho$. Here the domain wall forms along the short direction of the system and thus the limiting case is $\Theta_{b}^{(p)}(x_{\delta})$.

in the Pfaffian. Since we already need all four combinations of orientations, this procedure thus only exchanges the role of $Z_{e/o}^{+}$ and $Z_{e/o}^{-}$. Consequently, if we impose antiperiodic BCs $\sigma_{\ell,m} \equiv -\sigma_{\ell+L,m}$ in the perpendicular direction, the roles of $Z_{e}^{\pm}$ and $Z_{o}^{\pm}$ are exchanged. Tab. 1 shows which of the $Z_{e/o}^{\pm}$ contributes with a minus sign depending on the BCs.

Of course this directly transfers to the scaling functions, which are shown in Figs. 8, 9, and 10. Just like the case of periodic BCs in both directions, antiperiodic BCs (denoted $(a, a)$ in the superscript) in both directions impose a symmetry according to

$$\Theta_{\parallel}^{(a,a)}(x_{\parallel}, \rho) = \Theta_{\perp}^{(a,a)}(x_{\perp}, 1/\rho), \tag{5.1}$$

see Fig. 8, where the two combinations of periodic and anti-periodic BCs (denoted $(p, a)$ and $(a, p)$) are connected via

$$\Theta_{\parallel}^{(a,p)}(x_{\parallel}, \rho) = \Theta_{\perp}^{(p,a)}(x_{\perp}, 1/\rho), \tag{5.2a}$$

$$\Theta_{\parallel}^{(p,a)}(x_{\parallel}, \rho) = \Theta_{\perp}^{(a,p)}(x_{\perp}, 1/\rho), \tag{5.2b}$$

see Figs. 9 and 10. At criticality, i. e., for $x_{\delta} = 0$, all four scaling functions coincide with the results of Izmailian [60].

An important consequence of an antiperiodic boundary is the formation of a domain wall and thus a surface tension contribution $\sigma$ to the free energy. This contribution can be calculated as difference between the free energy of the system with antiperiodic and the one of the

Table 1: Sign of the contributions $Z_{e/o}^{\pm}$ according to the combination of periodic (p) and antiperiodic (a) boundary conditions in parallel and perpendicular directions.

| BCs | $Z_o^+$ | $Z_o^-$ | $Z_e^+$ | $Z_e^-$ |
|---|---|---|---|---|
| (p, p) | + | + | + | − |
| (p, a) | + | + | − | + |
| (a, p) | + | − | + | + |
| (a, a) | − | + | + | + |

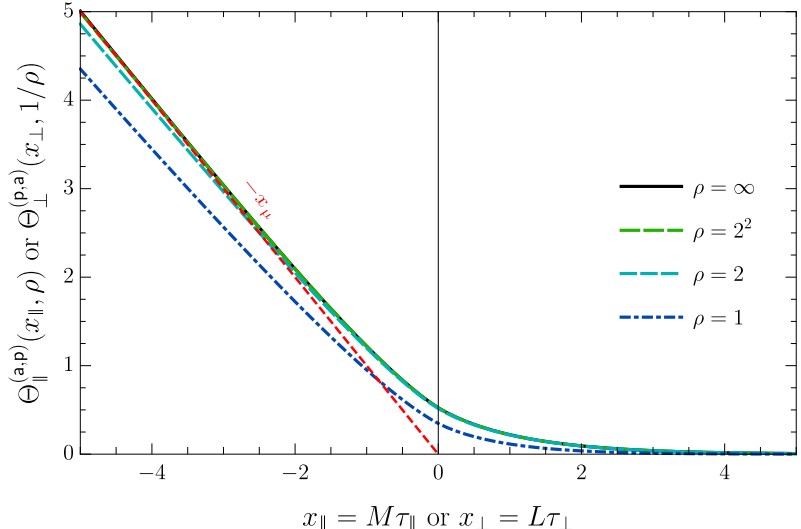

Figure 10: Scaling functions $\Theta_\parallel^{(a,p)}(x_\parallel, \rho)$ and due to the symmetry $\Theta_\perp^{(p,a)}(x_\perp, 1/\rho)$ for different values for the aspect ratio $\rho$. Here the domain wall forms along the long direction of the system and thus the limiting case is $\Sigma_b^{(a)}(x_\delta)$ with a linear divergence for $x_\delta \to -\infty$, see (5.6).

system with periodic boundaries in the same direction [82],

$$\sigma^{(a,p)}(L,M) = F^{(a,p)}(L,M) - F^{(p,p)}(L,M), \tag{5.3}$$

or alternatively as quotient of the corresponding partition functions. Thus any bulk or surface contribution cancels out and what is left is the free energy of the domain wall composed of the differences of the associated residual finite-size parts, which consequently fulfils the scaling *ansatz*

$$\sigma^{(a,p)}(L,M) \simeq \rho\, \Sigma_\parallel^{(a,p)}(x_\parallel, \rho), \tag{5.4}$$

with the scaling function

$$\Sigma_\parallel^{(a,p)}(x_\parallel, \rho) = \Theta_\parallel^{(a,p)}(x_\parallel, \rho) - \Theta_\parallel^{(p,p)}(x_\parallel, \rho). \tag{5.5}$$

Likewise, the surface tension and its scaling function are obtained for periodic/antiperiodic and antiperiodic/antiperiodic BCs. For all three cases the scaling functions diverge linearly in the corresponding scaling variable $x_\delta$, where $\delta$ might either be the parallel or the perpendicular direction, in the ordered phase for a growing length of the domain wall, as depicted in Figs. 8, 9, and 10. The limiting case for a dominating domain wall, i.e., antiperiodic boundaries in the long direction, reads

$$\Sigma_b^{(a)}(x_\delta) = \delta\Theta_b(x_\delta) - x_\delta H(-x_\delta), \tag{5.6}$$

where $\Sigma_b^{(a)}(x_\delta)$ is the bulk contribution to the surface tension, cf. [28], and $H(x)$ is the Heaviside step function with $H(x) \equiv \frac{d}{dx}\max\{0, x\}$.

# 6 Conclusion

We presented a systematic calculation of the universal free energy finite-size scaling functions for anisotropic Ising systems with translationally invariant boundary conditions. Therefore

we started with the commonly known dimer representation of the two-dimensional square lattice Ising model, which we recapitulated in some detail to show the connection between the Pfaffian representation and the boundary conditions of the underlying system. Using the coupling constant $z$ and its dual counterpart $t$ turned out to be beneficial for the calculation, as it compresses the formulas to a clearly arranged form. The calculation of the necessary Pfaffians of the original $4LM \times 4LM$ matrices was condensed down to a straightforward formalism for periodic and antiperiodic boundaries assuming translational invariance in one direction. Within this approach we implemented an anisotropic scaling theory and showed analytically that a coupling anisotropy can be absorbed into a generalised aspect ratio $\rho$. For the transition from the finite system to the scaling form we introduced the hyperbolic parametrisation of the scaling form of the Onsager dispersion relation, which is the scaling limit of the elliptic parametrisation of finite systems. The results are in perfect agreement with former calculations, but the method is far more general and we will use it to apply boundary fields to the open cylinder in a subsequent paper [28].

# Acknowledgements

We like to thank Malte Henkel for helpful discussions and inspirations, and Walter Selke for helpful comments.

**Funding information** This work was supported by the Deutsche Forschungsgemeinschaft through Grant HU 2303/1-1.

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
