# Peer review of "Anisotropic scaling of the two-dimensional Ising model I: the torus"

_SciPost Physics, doi:SciPost Phys. 7, 026 (2019)_

## Round 1 · Referee Report · Jacques H. H. Perk (Referee 1) · 2018-5-20

Strengths

1- This sequence of two papers has several new results

Weaknesses

1- Many citations are lacking

2- Lacks adequate comparisons with the literature

2- Presentation is hard to read and can be improved

Report

This paper is for major part also preparation for a second paper arXiv:1805.00369 and should, in my opinion, only be published if that paper is also to be published in this journal, preferentially back-to-back. I shall, therefore, also comment on the second paper.

The calculations in these papers appear sound, but the presentation is at times not as clear as it can be. The original setup is essentially the presentation of the book of McCoy and Wu, but with some differences that are not clearly discussed and it will not at all be easy for a novice in the field to read the paper. One difference is that McCoy and Wu add plus/minus signs from the start, whereas the manuscript starts with only have positive entries and antisymmetrizes late in (11). Even though it is a matter of doing steps in a different order, as the reader is asked to consult McCoy and Wu, one may expect more discussion.

The major objection to the current version is that several essential citations are lacking. For section 2.2 of the manuscript, with translation invariance in one direction, one would not just want to see an implicit citation to Chapter 14 of the book of McCoy and Wu given (chapter 4 was cited earlier), but also the original paper [B.M. McCoy and T.T. Wu, Phys. Rev. 176, Theory of a two-dimensional Ising model with random impurities. I. Thermodynamics, 631-643 (1968)] and several other ones. Two early ones are:

(1) H. Au-Yang and B.M. McCoy, Theory of layered Ising models: Thermodynamics, Phys. Rev. B 10, 886-891 (1974). This paper varies the vertical couplings periodically with period $n$ in one direction. This work has many citations, also for quantum chains, due to the formula for the critical point. Tracy by a limit of periodic approximants got the critical point for a Fibonacci Ising model [C.A. Tracy, Universality class of a Fibonacci Ising model, J. Stat. Phys., 51, 481-490 (1988)]. Extension to triangular and honeycomb lattices exist [W.F. Wolff and J. Zittartz, Layered Ising models on triangular and honeycomb lattices, Z. Phys.B 49, 139-148 (1982).] Au-Yang and Fisher calculated various scaling functions [H. Au-Yang and M.E. Fisher, Bounded and inhomogeneous Ising models. II. Specific-heat scaling function for a strip, Phys. Rev. B 11, 3469-3487 (1975). M.E. Fisher and H. Au-Yang, Critical wall perturbations and a local free energy functional, Physica A 101, 255-264 (1980). H. Au-Yang and M.E. Fisher, Wall effects in critical systems: Scaling in Ising model strips, Phys. Rev. B 21, 3956-3970 (1980). H. Au-Yang and M.E. Fisher, Criticality in alternating layered Ising models: I. Effects of connectivity and proximity, Phys. Rev. E 88, 032147. H. Au-Yang, Criticality in alternating layered Ising models: II. Exact scaling theory, Phys. Rev. E 88, 032148. D.B. Abraham and J. De Coninck, Description of phases in a film-thickening transition, J. Phys. A: Math. Gen. 16, L333 (1983).] This just to mention a few of the many citations.

(2) J.R. Hamm, Regularly spaced blocks of impurities in the Ising model: Critical temperature and specific heat, Phys. Rev. B 15, 5391-5411 (1977). This paper varies both the horizontal and vertical couplings periodically with period $n$ in one direction. It even studies the case with period 2 in the other direction, in some sense more general than done in the manuscript!

There are also many papers on finite size corrections for the isotropic Ising model [N.Sh. Izmailian and C.-K. Hu, Exact universal amplitude ratios for two-dimensional Ising models and a quantum spin chain, Phys. Rev. Lett. 86, 5160-5163 (2001). N.Sh. Izmailian, K.B. Oganesyan and C.-K. Hu, Exact finite-size corrections for the square-lattice Ising model with Brascamp-Kunz boundary conditions, Phys. Rev. E 65, 056132 (2002). E.V. Ivashkevich, N.Sh. Izmailian and C.-K. Hu, Kronecker's double series and exact asymptotic expansions for free models of statistical mechanics on torus, J. Phys. A: Math. Gen. 35, 5543-5561 (2002). N.Sh. Izmailian and C.-K. Hu, Finite-size effects for the Ising model on helical tori, Phys. Rev. E 76, 041118 (2007).]

An interesting numerical approach is by star-triangle transformations in the isotropic case for several lattices and boundary conditions that deserve comparison [X. Wu, N. Izmailian and W. Guo, Finite-size behavior of the critical Ising model on a rectangle with free boundaries, Phys. Rev. E 86, 041149 (2012), errata: 87, 019901(E) (2013). X. Wu, N. Izmailian and W. Guo, Shape-dependent finite-size effect of the critical two-dimensional Ising model on a triangular lattice, Phys. Rev. E 87, 022124 (2013). X. Wu, R. Zheng, N. Izmailian and W. Guo, Accurate expansions of internal energy and specific heat of critical two-dimensional Ising model with free boundaries, J. Stat. Phys. 155, 106-150 (2014).]

A must reference is: N.Sh. Izmailian, Finite-size effects for anisotropic 2D Ising model with various boundary conditions, J. Phys. A: Math. Theor. 45, 494009 (2012). See also: D.B. Abraham and N.M. {\v S}vraki{\'c}, Exact finite-size effects in surface tension, Phys. Rev. Lett. 56, 1172-1174 (1986). D.B. Abraham, L.F. Ko and N.M. {\v S}vraki{\'c}, Ising model with adjustable boundary conditions: Exact results for finite lattice mass gaps, Phys. Rev. Lett. 61, 2393-2396 (1988). D.B. Abraham, L.F. Ko and N.M. {\v S}vraki{\'c}, Transfer matrix spectrum for the finite-width Ising model with adjustable boundary conditions: Exact solution, J. Stat. Phys. 56, 563-587 (1989).

In the second paper, the authors discuss the case of two boundary fields. This was pioneered by Au-Yang [H. Au-Yang, Thermodynamics of an anisotropic boundary of a two-dimensional Ising model, J. Math. Phys. 14, 937-946 (1973).] This paper also has relevant citations.

There are many more citations, but this is what a brief citation search has given me. The authors should include most of these and possibly others at suitable places in their work and make comparisons with results in the literature of the last almost 50 years, as their work is seriously lacking in this.

Requested changes

1- Add many citations at suitable places

2- Make adequate comparisons of results with the literature

3- Improve the presentation to make paper more readable and to make results more accessible

  • validity: high
  • significance: high
  • originality: high
  • clarity: ok
  • formatting: good
  • grammar: good

Author:  Alfred Hucht  on 2019-06-23  [id 548]

(in reply to Report 1 by Jacques H. H. Perk on 2018-05-20)
Category:
remark
answer to question
reply to objection
correction

We thank Prof. Perk for his detailed report and changed the manuscript accordingly. Unfortunately, the whole procedure took much longer than expected due to several reasons. We kindly apologise for that delay.

We rewrote large parts of the manuscript, updated several figures, added a large number of citations (34), and changed the notation at many places on order to increase the readability and to simplify the calculation. We added the requested citations at the appropriate positions, where our results match already published work.

Concerning the criticism of citations to the layered Ising model, we have chosen not to include a complete list of papers concerning this model. Although it is translational invariant in one direction, the mentioned works cover only the infinitely large case and not finite lattices, not to speak of the case of a finite aspect ratio, and therefore only mimic true periodic behaviour in finite systems by repeating the couplings periodically. Consequently, these works consider no boundaries at all, and therefore allow for no predictions concerning residual free energies, finite-size scaling functions, and Casimir forces.

The first part of our work, on the other hand, is concerned with exactly this set up, i.e., finding the partition function for the periodic, finite system. The main difference can be located at criticality, as here the divergence of the correlation length is only truly restricted for a finite lattice; the periodic repetition of the couplings on the infinite lattice does not do so. Especially there is no self-interaction of the critical domains because of the critical percolation. That we first choosed to fix only one direction accounts for the will to present the most general formulation of our problem, in this case a handy transfer-matrix method for the periodic, finite Ising model in two dimensions.

Especially the paper by J.R. Hamm [J.R. Hamm, Regularly spaced blocks of impurities in the Ising model: Critical temperature and specific heat, Phys. Rev. B 15, 5391-5411 (1977)] is for our concerns in no sense more general, as it — again — only mimics true periodicity by a periodic variation of the couplings and not by true periodic boundary conditions.

We submitted the second part of this work to SciPost, together with an revised version of part 1, as suggested.

---

## Round 1 · Referee Report · Anonymous (Referee 2) · 2018-5-25

Strengths

  1. Gives a number of new results for the scaling form of the free energy of an Ising model on the torus, including a careful discussion of the Casimir effect.

Weaknesses

  1. The authors should state more carefully which results are being reviewed and which are claimed to be new.
  2. The existing (huge) literature should be better acknowledged, by citing original papers instead of textbooks and reviews.
  3. Some elements of the presentation could be improved.

Report

The paper begins by reviewing the derivation of the free energy of a two-dimensional square-lattice Ising model on the torus with various boundary conditions (periodic and antiperiodic along the two principal directions), using the dimer representation. This review is useful and contains some slight changes with respect to the existing literature. However, this part would be stronger if the authors accounted more precisely for those changes and cited the existing literature more carefully (instead of giving general citations to the book by McCoy and Wu). There is also a number of places where the wording is unclear, or English words are not used correctly (a few examples: "exemplary" in the caption to figure 2, "covered by the two matrices" before (7b), "emulated" on page 7).

The more interesting part of the manuscript starts in section 3 with a careful discussion of the scaling form of the free energy and the extraction of various kinds of Casimir effects. However, the authors should make a greater effort to state which results can be found elsewhere (in some form) and which ones are genuinely new. In particular, similar investigations have been performed in several papers by Izmailian and coworkers, none of which are cited here.

The paper would gain in interest if it contained also the treatment of boundary fields, as announced towards the end, but if the authors insist on publishing this separately I do not have any objections.

Requested changes

  1. Cite original papers instead of general textbooks, and add missing references.
  2. Correct unclear or erroneous wordings.
  3. Make clear claims which results are genuinely new.

  • validity: high
  • significance: high
  • originality: good
  • clarity: good
  • formatting: excellent
  • grammar: good

Author:  Alfred Hucht  on 2019-06-23  [id 547]

(in reply to Report 2 on 2018-05-25)
Category:
remark
answer to question
correction

We thank the referee for her/his report and changed the manuscript accordingly. Unfortunately, the whole procedure took much longer than expected due to several reasons. We kindly apologise for that delay.

We rewrote large parts of the manuscript, updated several figures, added a large number of citations (34), and changed the notation at many places on order to increase the readability and to simplify the calculation. We added the requested citations at the appropriate positions, where our results match already published work.

We kept the second part of this work separately and submitted it to SciPost, together with an revised version of part 1, as suggested by the first referee JHH Perk.

---

## Round 2 · Referee Report · Anonymous (Referee 2) · 2019-7-12

Strengths

1- The authors obtain new results on the anisotropic Ising model in a toroidal geometry. 2- The simplifications with respect to previous methods should allow the authors to treat more general cases in the future.

Weaknesses

1- Prior to acceptance, the authors would need to correct a few minor details, mainly having to do with English usage and spelling.

Report

The authors have responded very thoroughly to the comments on an earlier version. The paper can now be accepted, after the correction of a few minor details.

Requested changes

1- My list of changes turns out to be a (proper) subset of those requested by the first referee, so the authors can refer to his or her list.

---

## Round 2 · Referee Report · Anonymous (Referee 3) · 2019-7-12

Strengths

Paper gives new results on finite-size scaling limits of the anisotropic Ising model on a torus.

Weaknesses

See requested changes.

Report

The resubmitted manuscript has adequate citations and new results added and, as far as I can see, these results are correct. I have gone through a lot of the calculations and even though I would say some things differently, the authors should have the freedom to say those things their own way. I recommend publication after the authors clean up several minor issues, including both confusing and trivial spelling errors. It should not take them much time. This paper can now stand on its own after new results were added on the finite-size scaling functions for anisotropic Ising on a torus.

Requested changes

First, the word "according" appears many times in the manuscript, where other words like "corresponding", "associated" or "related" fit better in the context and make the meaning clearer.

Page 2, line 16 about: I am not comfortable with the "Only recently ..." statement. The "direct connection" does not occur early in the two papers. Also, there are earlier instances where dimer and spinor calculations for Ising meet, especially for correlation functions. Rolling up the model row-by-row and applying Fourier transform, 2x2 matrices already showed up very early in dimer model free-energy calculations also. A sentence like "It is very interesting that [9] using spinors and [18,19] using dimers end up with the same 2x2 matrices ..." 0r similar seems more appropriate.

Page 2, 3rd line paragraph 3: "spatial" is the preferred spelling; "spacial" is improper usage that has become accepted.

Page 3, line 4: "are" needs to be deleted.

Page 3, paragraph 2: The opening "As" may have to be omitted and the comma before "furthermore" may then be replaced by ";".

Page 4, section 2, line 3: It is fair to say here that McCoy and Wu pioneered this in Chapter XIV of their book based on Phys. Rev. 176, 631 (1968). The generalization here follows their method with the horizontal couplings also varying from row to row.

Pages 5-8: The writing is not clear to people not familiar with the dimer method. They would be required to consult the McCoy-Wu book or similar. It would help these readers, if they were told to consult chapter IV and section 2 of Chapter V of that book for a rather extensive treatment of the method.

Page 13, lines 4 and 5: extend (verb) should become extent (noun).

Page 15, line above (3.17): "scaling functions" should be plural (or "consists" singular).

Page 16: Minor remark (optional): (4.3) is (anti)cyclic, so the determinant can also be found by diagonalizing with Fourier similarity transform.

Page 17, (4.9): At this point it may also be good to remind people of (2.3) for $Z_0^{(p,p)}$. They would have to really seek for it.

Page 20, line 1: "throughout" is one word.

Page 23, bottom line: Replace "Eventually" by "Finally".

Page 24, line below (4.37): Replace "for most" by "foremost".

Page 25, bottom line: Remove comma in "see, Fig. 8".

Page 27, 2 lines below (5.6): Replacing $\partial_x$ by $d/dx$ is clearer.

Page 27, 2 lines from bottom: "Therefor" should be "Therefore,".

Page 28, middle: "the subsequent part of this paper [28]" sound strange. Could be changed to "a subsequent paper [28]", "a companion paper [28]", "a later paper [28]", or similar.

---

## Round 2 · Author Response

Dear editors,

we thank both referees for their reports and changed the manuscript accordingly. Furthermore, we submitted the second part of the work [arXiv:1805.00369v2] to SciPost, as suggested by the first referee JHH Perk. We rewrote large parts of the manuscript, updated several figures, and changed the notation at many places on order to increase the readability and simplify the calculation.

---

## Round 2 · List of Changes

o) Equations are now numbered by section
o) Added text and references to Introduction
o) Added several references to the other sections
o) Added a minus sign to H (2.8), and updated the following equations accordingly, such that antiperiodic boundary conditions in the Ising model get the subscript "-".
o) Changed $J_{\beta}^{-}$ to $-J_{\beta}^{-}$ in (2.23)
o) Explicitly defined the dual (2.24)
o) Renamed matrices $\Delta,\Sigma$ to $K^{\pm}$
o) Removed Fig. 4
o) Moved (46a) to (3.2)
o) Simplified (3.25)
o) Removed the incorrect paragraph after (3.25)
o) Simplified (4.35)
o) Changed subsection 4.3 to be section 5

---

## Round 3 · Author Response

Dear editors,

we thank both referees for their comprehensive reports and changed the manuscript accordingly. Especially report 1 was really helpful and increased the readability of the manuscript. We made all requested changes as recommended, see below, with one exception: The sentence "Only recently..." on page 2, where we commented on the connection between the transfer matrices derived within the dimer method [18,19] and the spinors in [9], was reformulated and extended differently from the referee's recommendations. It now reads:

"Only quite recently, a connection between these two methods was established [18,19], which reduces the Pfaffians emerging in the dimer method to matrices corresponding to the spinor picture even for arbitrary couplings, therefore preserving the possibility to apply arbitrary boundary conditions in both directions [20]. Note that this correspondence goes beyond the simpler case of translational invariant couplings, where both methods are known to lead to the same 2 × 2 matrices."

---

## Round 3 · List of Changes

As requested in Report 1:
o) Changed the word "according" throughout the manuscript.
o) Rewrote the statement "Only recently..." on page 2 to be more precise.
o) Reformulated the sentence page 3, line 4 "Other experiments...".
o) Changed "As the..." to "The..." on page 3, par 2.
o) Added reference to McCoy & Wu [16] on page 4, sec 2. Note that this reference was already given on page 3.
o) Added reference to McCoy & Wu [17] to chapter "Dimers".
o) We fixed all typos: "spacial" -> "spatial" on p.2, "extend" -> "extent" on p.13, etc., see "Requested changes".
o) Changed the phrase "the subsequent part of this paper" to "a subsequent paper" as recommended.

Other changes:
o) Changed $\partial_{\Phi}$ to $\frac{\partial}{\partial\Phi}$ in (4.24).
o) Updated references.

---

## Editorial Decision

published